# MORE: MULTI-OBJECTIVE ADVERSARIAL ATTACKS ON SPEECH RECOGNITION

## ABSTRACT

The emergence of large-scale automatic speech recognition (ASR) models such as Whisper has greatly expanded their adoption across diverse real-world applications. Ensuring robustness against even minor input perturbations is therefore critical for maintaining reliable performance in real-time environments. While prior work has mainly examined accuracy degradation under adversarial attacks, robustness with respect to efficiency remains largely unexplored. This narrow focus provides only a partial understanding of ASR model vulnerabilities. To address this gap, we conduct a comprehensive study of ASR robustness under multiple attack scenarios. We introduce MORE, a multi-objective repetitive doubling encouragement attack, which jointly degrades recognition accuracy and inference efficiency through a hierarchical staged repulsion–anchoring mechanism. Specifically, we reformulate multi-objective adversarial optimization into a hierarchical framework that sequentially achieves the dual objectives. To further amplify effectiveness, we propose a novel repetitive encouragement doubling objective (REDO) that induces duplicative text generation by maintaining accuracy degradation and periodically doubling the predicted sequence length. Overall, MORE compels ASR models to produce incorrect transcriptions at a substantially higher computational cost, triggered by a single adversarial input. Experiments show that MORE consistently yields significantly longer transcriptions while maintaining high word error rates compared to existing baselines, underscoring its effectiveness in multi-objective adversarial attack.

## 1 INTRODUCTION

Automatic speech recognition (ASR) models, exemplified by the Whisper family (Radford et al., 2023), have become integral to a wide range of applications, including virtual assistants, real-time subtitling, clinical documentation, and spoken navigation (Gao & Chen, 2024). Despite their success, the reliability of these systems in practical deployments remains fragile: even small adversarial perturbations can substantially degrade recognition accuracy or disrupt inference efficiency—for instance, by causing misinterpretation of user commands or inducing denial-of-service behaviors. These vulnerabilities underscore the need for a systematic examination of ASR robustness across both accuracy and efficiency, which is essential for ensuring dependable performance in real-world, time-sensitive environments.

Most prior work has been dedicated to accuracy robustness under adversarial attacks (Raina et al., 2024; Raina & Gales, 2024; Olivier & Raj, 2022b; Madry et al., 2018a; Dong et al., 2018; Wang & He, 2021; Gao et al., 2024). While these efforts help understanding ASR model accuracy vulnerabilities, the efficiency robustness of ASR models, and their ability to maintain real-time inference under adversarial conditions remain largely unexplored. Such efficiency is critical, as adversaries can exploit it to degrade system responsiveness, e.g., causing systems to output unnaturally long transcripts, severely impacting usability and causing the inference process to be excessively time-consuming. Therefore, enhancing and evaluating the efficiency robustness of ASR models is crucial to ensure their practicality in real-time, user-facing systems.

As efficiency robustness plays a pivotal role in the real-world applicability of deep learning models, there is a growing need to systematically assess it. Recent research has proposed adversarial attack methods to evaluate efficiency robustness in various domains, including computer vision (Li

et al., 2023b; Chen et al., 2022b), machine translation (Chen et al., 2022a), natural language processing (Chen et al., 2023; Li et al., 2023a; Ebrahimi et al., 2018; Li et al., 2019), and speech generation models (Gao et al., 2025). However, research on the efficiency robustness of ASR models under attacks remains critically scarce, with SlothSpeech (Haque et al., 2023) standing as the only known effort. Yet, SlothSpeech does not consider the impact of efficiency attacks on accuracy and does not systematically explore adversarial output patterns. This leaves the efficiency dimension of ASR robustness insufficiently examined and calls for further investigation.

Nevertheless, the robustness of both accuracy and efficiency in ASR models still lags considerably behind human speech recognition performance (Haque et al., 2023; Gao et al., 2024). This stark disparity underscores the need for a more holistic investigation into the vulnerabilities of these models. In this paper, we conduct a comprehensive study of the robustness of the Whisper family, a set of representative large-scale ASR models, with respect to both accuracy and efficiency.

To this end, we propose a novel **M**ulti-**O**bjective **R**epetitive Doubling **E**ncouragement attack approach (**MORE**) that simultaneously targets both accuracy and efficiency vulnerabilities. Unlike prior attacks that optimize a single objective, MORE incorporates a multi-objective *repulsion–anchoring* optimization strategy that unifies accuracy-based and efficiency-based adversarial attacks within a single network. Motivated by natural human speech repetitions and repetitive decoding loops observed in transformer-based models Xu et al. (2022), we introduce a repetitive encouragement doubling objective (REDO) that promotes duplicative text pattern generation periodically by maintaining accuracy degradation in producing elongated transcriptions. An asymmetric interleaving mechanism further reinforces periodic context doubling while an EOS suppression objective discourages early termination.

The contributions of this paper include: (a) this paper presents the first unified attack approach that jointly targets both accuracy and efficiency robustness against large-scale ASR models via a multi-objective *repulsion-anchoring* optimization strategy; (b) we propose **REDO**, which bridges efficiency with accuracy gradients via guiding the accuracy-modified gradients towards repetitive elongated semantic contexts, thereby inducing incorrect yet extended transcriptions; and (c) we provide a comprehensive comparative study of diverse attack methods with insightful findings to balance accuracy and efficiency degradation. Extensive experiments demonstrate that the proposed **MORE** consistently outperforms all baselines in producing longer transcriptions while maintaining strong accuracy attack performance.

## 2 RELATED WORK

**Adversarial Attacks on Speech Recognition.** Automatic speech recognition has been extensively studied regarding its vulnerability to attacks. These attacks primarily seek to degrade recognition accuracy by introducing typically subtle perturbations into speech inputs, thereby compromising transcription accuracy (Haque et al., 2023; Olivier & Raj, 2022a; Schönherr et al., 2018; Wang et al., 2022; Zhang et al., 2022; Ge et al., 2023). Notable examples include attacks in the MFCC feature domain (Vaidya et al., 2015; Dan Iter, 2017), targeted attacks designed to trigger specific commands (Carlini et al., 2016), and perturbations constrained to ultrasonic frequency bands (e.g., DolphinAttack Zhang et al. (2017)). Most prior works on attacking ASR have concentrated on traditional architectures, such as CNN or Kaldi-based systems (Wang et al., 2020), with limited exploration into modern large-scale transformer-based ASR models.

Recent advances in ASR have been driven by the emergence of large-scale models, notably OpenAI Whisper (Radford et al., 2023), a transformer-based encoder-decoder architecture trained on large-scale datasets (680K hours of data), demonstrating greater robustness and generalization across diverse speech scenarios. Consequently, there has been an increasing research interest in evaluating the adversarial robustness of Whisper, particularly focusing on accuracy-oriented attacks. Such efforts include universal attacks (Raina et al., 2024; Raina & Gales, 2024), targeted Carlini&Wagner (CW) attacks (Olivier & Raj, 2022b) and gradient-based methods, i.e., projected gradient descent (PGD) (Madry et al., 2018a), momentum iterative fast gradient sign method (MI-FGSM) (Dong et al., 2018), variance-tuned momentum iterative fast gradient sign method (VMI-FGSM) (Wang & He, 2021), as well as speech-aware adversarial attacks (Gao et al., 2024). However, most existing approaches focus only on accuracy robustness and overlook vulnerabilities in inference efficiency, which can be exploited through decoding manipulation. SlothSpeech (Haque et al., 2023) represents

the only prior efficiency-focused attack in ASR, but it does not jointly consider accuracy degradation or structured repetition, limiting its ability to assess multi-dimensional robustness.

**Motivations and Applications.**    Different from previous attacks, our proposed MORE systematically evaluates and undermines both accuracy and efficiency within a single adversarial network, offering a comprehensive understanding of large-scale ASR model's vulnerabilities that previous single-objective methods cannot provide. The significance of studying the adversarial robustness of ASR models, particularly Whisper, is amplified by their potential deployment in hate speech moderation (MacAvaney et al., 2019; Wu & Bhandary, 2020) and private speech data protection. Practically, our proposed MORE can be applied to distort the transcription of harmful or private speech, preventing ASR systems from reliably converting such content into readable text. By inducing incorrect and excessively long transcriptions, MORE exposes decoding weaknesses that are not revealed by accuracy-only attacks, offering a more comprehensive view of ASR vulnerability.

## 3    MORE

### 3.1    PROBLEM FORMULATION

**Victim model.**    We consider a raw speech input represented as a sequence $X = [x_1, x_2, \ldots, x_T]$. Its corresponding ground-truth transcription is a sequence of text tokens $Y = [y_1, y_2, \ldots, y_L]$. The target ASR model is denoted by a function $f(\cdot)$ that maps a speech sequence to a predicted transcription, i.e., $f(X) = \hat{Y}$. The model vocabulary is denoted by $V$, and EOS $\in V$ is the end-of-sequence token. Our objective is to construct an adversarial perturbation $\delta$ such that the perturbed input $X + \delta$ triggers harmful behavior during decoding.

**Attack objective.**    Most existing adversarial attacks on ASR aim solely to maximize transcription error. However, practically disruptive attacks must also degrade inference efficiency, especially in real-time ASR systems where excessive decoding time can break user interactions. We therefore formulate a dual-objective optimization targeting both transcription accuracy and computational efficiency:

$$S = \arg \max_{\delta \in \Delta_\infty(\epsilon)} \left( \text{WER}\big(f(X + \delta), Y\big), \big|f(X + \delta)\big| \right) \quad (1)$$

where $\text{WER}(\cdot)$ denotes the word error rate and $|f(\cdot)|$ denotes the length of the predicted sequence. This formulation explicitly seeks perturbations that (i) increase transcription error relative to the ground truth and (ii) induce excessively long outputs, thereby amplifying computational overhead.

**Perturbation constraint.**    We impose both energy- and peak-based constraints for imperceptibility. A standard measure is the signal-to-noise ratio (SNR), which compares the energy of the signal and the perturbation:

$$\text{SNR}_{\text{dB}} = 20 \log_{10} \left( \frac{\|X\|_2}{\|\delta\|_2} \right). \quad (2)$$

While SNR constrains overall perturbation energy, it may still allow short, high-amplitude distortions. To avoid this, we additionally bound the perturbation's peak amplitude using the $\ell_\infty$ norm:

$$\Delta = \{\delta \mid \|\delta\|_\infty \leq \epsilon\}, \quad \text{where} \quad \epsilon = \frac{\|X\|_\infty}{\text{SNR}}. \quad (3)$$

This $\ell_\infty$ constraint ensures that no single sample deviates excessively, which aligns with psychoacoustic masking principles. The adversarial example is thus defined as $X_{\text{adv}} = X + \delta$ with $\delta \in \Delta$.

### 3.2    DESIGN OVERVIEW AND MOTIVATIONS

The proposed **MORE** attack is motivated by the autoregressive nature of ASR models and the different optimization dynamics of our two goals: reducing transcription accuracy and prolonging decoding for efficiency degradation, as illustrated in Fig. 1. In autoregressive models, each predicted token influences all future predictions, with the end-of-sentence (EOS) token being particularly sensitive; small perturbations to its logits can drastically alter when decoding stops (Raina et al., 2024; Olivier & Raj, 2022b), but it receives sparse gradient signal compared with ordinary tokens. The accuracy

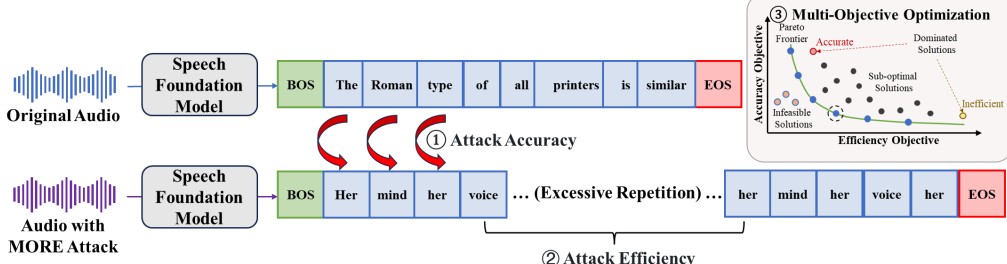

Figure 1: Overview of our proposed **m**ulti-**o**bjective **r**epetitive **e**ncouragement doubling (MORE) adversarial attack method.

attack objective, in contrast, distributes across many token positions, encouraging mis-transcriptions and resulting in a relatively large feasible adversarial set, as many incorrect transcripts are possible. The efficiency attack objective, however, mainly targets the non-stopping behavior associated with a single EOS token, where gradients are narrowly concentrated and typically smaller in magnitude compared with those of the broader accuracy objective. Because accuracy gradients are broad and efficiency gradients are sharp and concentrated, combining them in a single-step optimization often causes one objective to dominate. This makes direct multi-objective optimization unstable.

To address this, our proposed **MORE** uses a hierarchical two-stage strategy consisting of a repulsion stage for accuracy degradation and an anchoring stage for efficiency degradation. The repulsion stage forces the model away from the correct transcription. The anchoring stage then exploits remaining degrees of freedom to extend decoding.

Formally, we approximate the hierarchical formulation in Eq. 4 using a two-stage repulsion-anchoring method. In the *repulsion* stage, we maximize a differentiable proxy of WER. In the *anchoring* stage, we extend the decoded sequence length while preserving the high error rate obtained in the *repulsion* stage. The following sections describe the optimization procedure for each in detail.

$$S = \arg \max_{\delta \in \Delta_\infty(\epsilon)} \text{WER}\big(f(X + \delta), Y\big), \quad \delta^* = \arg \max_{\delta \in S} \big|f(X + \delta)\big|, \tag{4}$$

This staged hierarchical design avoids forcing accuracy and efficiency gradients to compete simultaneously and provides a stable optimization path.

### 3.3 REPULSION STAGE: ACCURACY ATTACK

The first pillar of our **MORE** attack is the *repulsion* stage, which focuses on degrading transcription accuracy. The repulsion stage applies a standard gradient-based accuracy-degradation attack using cross-entropy (CE) as a differentiable proxy for WER. Minimizing negative CE reduces the probability of ground-truth tokens, pushing the model toward incorrect outputs and increasing WER.

The accuracy attack loss is formulated as:

$$\mathcal{L}_{\text{acc}} = -\text{CE}\big(f(X + \delta), Y\big). \tag{5}$$

Taking gradient changing steps *w.r.t.* this loss function encourages the ASR model to output incorrect tokens, which directly correlates with an increase in the ultimate WER. This serves as the initial "destabilization" repulsion stage of our attack to destabilize the decoding trajectory and prepare the model for the subsequent efficiency attack.

### 3.4 ANCHORING STAGE: EFFICIENCY ATTACK

Complementing the accuracy degradation, MORE's efficiency attack targets the now-vulnerable model by anchoring it to generate excessively long and computationally expensive transcriptions. This *anchoring* stage is accomplished through the two components below.

**EOS suppression.** Decoding normally terminates when EOS is predicted. By penalizing the probability of this token, we can deceive the model into prolonging the decoding process indefinitely, often resulting in the generation of irrelevant or meaningless tokens. Penalizing only the EOS token is insufficient since its probability is typically dominant at the final decoding step. To enhance attack effectiveness, we reduce the likelihood of the EOS token. In addition, we increase the probability of the competing token with the second-largest likelihood. Reinforcing this alternative token not only diminishes EOS dominance but also guides the model toward continued generation. Therefore, the EOS-suppression loss is formulated as:

$$\mathcal{L}_{\text{EOS}} = P_L^{\text{EOS}} - P_L^z, \tag{6}$$

where $P_L^{\text{EOS}}$ is the probability that the model emits EOS at the final position $L$. z is the token with the second largest probabilities at position $L$:

$$z = \arg \max_{v \in V \setminus \{\text{EOS}\}} P_L^v \tag{7}$$

We denote $L$ as the output sequence length, and $P_L^v$ is the model's predicted probability of token $v$ being produced at output position $L$. This dual adjustment ensures the model is discouraged from selecting EOS while being nudged toward an alternative continuation, thereby minimizing this loss reduces EOS dominance and favors continuation tokens, which in turn prolongs decoding.

**Repetitive Encouragement Doubling Objective (REDO).** While effective, simple EOS suppression can lead to unstable optimization or low-confidence, random outputs. To introduce a more structured and potent method for sequence elongation, we propose a novel *repetitive encouragement doubling objective (REDO),* inspired by repetition loops observed in transformer models (Xu et al., 2022) and natural speech disfluencies. Transformer models are known to enter self-reinforcing repetition loops where once a sentence with high generation probability is produced, the model tends to reproduce it in subsequent steps, as its presence in the context further boosts its likelihood of being selected again (Xu et al., 2022). This recursive amplification leads to a self-sustaining loop of repetition, wherein repeated sentences reinforce their own future generation by dominating the context.

Our REDO leverages this mechanism to force long structured repetitions, thereby reliably increasing sequence length, as demonstrated in Figure 1. At each period, REDO constructs a duplicated version of the earlier decoded segment and uses CE to encourage the model to reproduce the extended sequence consistently. This produces stable semantic repetition and much longer sequences than EOS suppression alone.

Specifically, given an initial decoding output $\hat{Y}$, we construct a new target sequence $\bar{Y}$ that contains a repeated segment. We then force the model to predict this new, longer sequence using a cross-entropy objective. The target sequence $\bar{Y}_i$ for step $i$ is constructed as:

$$\bar{Y}_i = \hat{y}_{\lfloor \frac{i}{D} \rfloor}[1:L-1] \,\|\, \hat{y}_{\lfloor \frac{i}{D} \rfloor}[1:L-1], \tag{8}$$

where $L$ is the length of target sequence length at step $\lfloor \frac{i}{D} \rfloor$, $D$ is the the doubling period, controlling how frequently the sequence is duplicated. The floor function $\lfloor \frac{i}{D} \rfloor$ ensures that the repeated segment remains fixed within each interval of D steps, only updating once every D steps. This periodic repetition creates stable semantic loops that encourage longer and more redundant model outputs. The doubling loss REDO is:

$$\mathcal{L}_{\text{REDO}} = \text{CE}(f(X + \delta), \bar{Y}_i). \tag{9}$$

For a concrete example, if the target sequence for attacking step 0 is $\bar{Y} = [y_1, y_2, y_3, \text{EOS}]$, the target sequence for attacking step 0 to 9 should be $\bar{Y} = [y_1, y_2, y_3, y_1, y_2, y_3]$ with doubling the regular tokens and strictly eliminating the EOS token. This loss explicitly guides the model to produce periodic, repeated segments, which serve to rapidly and reliably inflate the output token count while maintaining a degree of linguistic structure, making the attack more potent. Finally, for efficiency attack, the loss is formulated as $\mathcal{L}_{\text{eff}} = \mathcal{L}_{\text{REDO}} + \mathcal{L}_{\text{EOS}}$.

**Asymmetric interleaving.** Applying a single-stage long-repeated target for attack can destabilize gradient optimization with the long-horizon optimization difficulties (Bengio et al., 2009; 2015; Madry et al., 2018b). To mitigate this, REDO breaks the long-horizon repetition task into a sequence of easier subproblems, yielding smoother optimization than trying to force a single-stage long-target

---

**Algorithm 1 MORE**: Hierarchical Attack with Curriculum Interleaved Efficiency Losses

---

1: **Input:** Original audio $X$, true transcript $Y$, $\ell_\infty$ radius $\epsilon$, step size $\alpha$, total steps $K$, accuracy steps $K_a$, doubling period $D$, ASR model $f(\cdot)$.
2: **Output:** Adversarial perturbation $\delta$
3: Initialize $\delta \leftarrow 0$
4: **Repulsion Stage: Accuracy (steps $1 \ldots K_a$)**
5: **for** $i = 1$ to $K_a$ **do**
6:     Compute $\mathcal{L} \leftarrow \mathcal{L}_{\text{acc}}$ by Eq. 5
7:     $\delta \leftarrow \text{clip}_{[-\epsilon,\epsilon]}\big(\delta - \alpha \cdot \text{sign}(\nabla_\delta \mathcal{L})\big)$
8: **end for**
9: **Anchoring Stage: Efficiency (steps $K_a+1 \ldots K$)**
10: **for** $i = K_a + 1$ to $K$ **do**
11:     $s \leftarrow i - K_a$
12:     **if** $(s-1) \bmod D = 0$ **then**
13:         Calculate $\bar{Y}$ by Eq. 8
14:     **end if**
15:     Compute $\mathcal{L}_{\text{EOS}}$ by Eq. 6
16:     Compute $\mathcal{L}_{\text{RDEO}}$ by Eq. 9
17:     $\mathcal{L} \leftarrow \mathcal{L}_{\text{EOS}} + \mathcal{L}_{REDO}$
18:     $\delta \leftarrow \text{clip}_{[-\epsilon,\epsilon]}\big(\delta - \alpha \cdot \text{sign}(\nabla_\delta \mathcal{L})\big)$
19: **end for**
20: **return** $\delta$

---

| Attack Methods | Whisper-tiny | | Whisper-base | | Whisper-small | | Whisper-medium | | Whisper-large | |
|---|---|---|---|---|---|---|---|---|---|---|
| | WER | length | WER | length | WER | length | WER | length | WER | length |
| **LibriSpeech Dataset** | | | | | | | | | | |
| clean | 6.66 | 21.84 | 4.90 | 21.77 | 3.64 | 21.84 | 2.99 | 21.84 | 3.01 | 21.84 |
| PGD | 93.17 | 35.02 | 88.73 | 31.65 | 75.23 | 27.93 | 64.77 | 26.94 | 33.33 | 21.80 |
| SlothSpeech | 46.80 | 119.38 | 54.63 | 156.07 | 38.25 | 110.40 | 31.09 | 81.75 | 34.21 | 79.78 |
| SAGO | 93.19 | 31.93 | 88.23 | 33.85 | 74.67 | 27.45 | 62.46 | 25.97 | 30.26 | 21.12 |
| VMI-FGSM | 87.91 | 32.39 | 93.47 | 33.90 | 73.84 | 27.34 | 60.74 | 25.38 | 29.77 | 20.79 |
| MI-FGSM | 93.32 | 34.38 | 87.57 | 34.44 | 74.18 | 27.07 | 61.16 | 25.70 | 34.27 | 21.40 |
| MORE | 91.01 | **296.28** | 88.42 | **300.13** | 74.28 | **213.94** | 64.04 | **234.25** | 53.72 | **301.47** |
| **LJ-Speech Dataset** | | | | | | | | | | |
| clean | 5.34 | 18.55 | 3.77 | 18.65 | 3.33 | 18.64 | 3.33 | 18.55 | 3.36 | 18.55 |
| PGD | 93.63 | 30.60 | 90.23 | 29.98 | 77.08 | 22.67 | 65.07 | 22.71 | 26.54 | 18.55 |
| SlothSpeech | 47.10 | 116.05 | 59.98 | 187.9 | 32.52 | 65.85 | 22.23 | 101.11 | 21.14 | 35.33 |
| SAGO | 89.53 | 31.63 | 84.46 | 26.69 | 72.08 | 22.31 | 59.08 | 21.80 | 21.03 | 19.11 |
| VMI-FGSM | 90.26 | 29.75 | 93.85 | 30.97 | 75.27 | 21.90 | 60.56 | 21.10 | 23.00 | 18.13 |
| MI-FGSM | 93.98 | 32.48 | 89.14 | 28.19 | 74.55 | 21.80 | 59.10 | 22.31 | 28.02 | 18.37 |
| MORE | 90.85 | **296.66** | 89.53 | **313.64** | 74.33 | **208.51** | 58.86 | **204.18** | 43.13 | **231.52** |

Table 1: Comparison of average recognition accuracy (WER%) and average transcribed text token length of various attack methods on the LibriSpeech and LJ-Speech datasets at an SNR of 35 dB. The reported accuracy and token length are averaged over 500 utterances for each dataset. 'Clean' denotes performance on the original, unperturbed speech. Note that higher WER and longer transcribed token length indicate a more successful attack.

objective. REDO is therefore formulated as a stepwise, curriculum-style attack that progressively optimizes for longer repeated outputs. Concretely, we *interleave*: for step $s$ maintain the repeated target fixed when $s \bmod D \neq 0$ and extend it to the next longer form when $s \bmod D = 0$. This "periodic booster" concentrates high-variance, long-horizon REDO updates sparsely while using frequent short-horizon updates to stabilize learning, and is distinct from summing losses or applying dense REDO updates at every step.

**Algorithm Details.** We integrate these components into a unified hierarchical procedure (Algorithm 1) to the dual-objective problem. We provide a detailed analysis in Appendix B. In particular, Appendix B characterizes how the computational cost of both the repulsion (accuracy-degradation) and anchoring (REDO-based repetition) stages scales with model depth, width, and the REDO-induced growth in output length. We additionally provide FLOPs analysis in Appendix C.

| | Whisper-tiny | | Whisper-base | | Whisper-small | | Whisper-medium | | Whisper-large | |
|---|---|---|---|---|---|---|---|---|---|---|
| Attack Methods | WER | length | WER | length | WER | length | WER | length | WER | length |
| **LibriSpeech Dataset** | | | | | | | | | | |
| clean | 6.66 | 21.84 | 3.77 | 21.77 | 3.64 | 21.84 | 2.99 | 21.84 | 3.36 | 21.84 |
| PGD | 96.22 | 35.30 | 93.90 | 33.21 | 85.83 | 31.40 | 78.77 | 28.87 | 47.93 | 21.92 |
| SlothSpeech | 60.06 | 123.93 | 69.33 | 152.82 | 56.45 | 102.15 | 41.75 | 77.43 | 48.34 | 78.65 |
| SAGO | 95.64 | 34.44 | 93.72 | 32.95 | 83.94 | 28.28 | 74.67 | 27.00 | 45.67 | 21.75 |
| VMI-FGSM | 93.23 | 37.53 | 96.08 | 37.72 | 83.15 | 28.06 | 73.66 | 27.02 | 42.75 | 22.07 |
| MI-FGSM | 96.18 | 34.37 | 93.08 | 33.22 | 83.49 | 28.37 | 72.52 | 27.54 | 45.35 | 22.11 |
| MORE | 94.73 | **300.79** | 93.70 | **324.05** | 86.15 | **238.52** | 77.84 | **202.34** | 60.90 | **277.65** |
| **LJ-Speech Dataset** | | | | | | | | | | |
| Clean | 5.34 | 18.55 | 4.90 | 18.65 | 3.33 | 18.64 | 3.33 | 18.55 | 3.01 | 18.55 |
| PGD | 96.56 | 31.79 | 94.46 | 30.50 | 86.08 | 25.69 | 79.45 | 24.42 | 42.87 | 19.97 |
| SlothSpeech | 61.62 | 123.58 | 71.51 | 187.87 | 51.33 | 76.59 | 38.33 | 103.33 | 35.74 | 37.50 |
| SAGO | 91.23 | 29.77 | 87.85 | 29.67 | 80.80 | 22.58 | 71.50 | 22.65 | 33.22 | 19.08 |
| VMI-FGSM | 93.46 | 28.67 | 96.57 | 33.55 | 84.15 | 24.15 | 73.05 | 22.16 | 36.28 | 19.31 |
| MI-FGSM | 96.15 | 31.04 | 93.37 | 29.18 | 83.97 | 23.37 | 73.15 | 22.47 | 42.71 | 18.21 |
| MORE | 94.80 | **326.62** | 94.08 | **310.57** | 85.49 | **222.04** | 75.01 | **229.66** | 54.13 | **229.08** |

Table 2: Comparison of average recognition accuracy (WER%) and average transcribed text token length of various attack methods on the LibriSpeech and LJ-Speech datasets at an SNR of 30 dB.

## 4 EXPERIMENTS

**Datasets.** We utilize two widely used ASR datasets from HuggingFace, LibriSpeech Panayotov et al. (2015) and LJ-Speech Ito & Johnson (2017), and evaluate the first 500 utterances from each (LJ-Speech and the test-clean subset of LibriSpeech), with all audio resampled to 16,000 Hz.

**Threat Model.** We conduct white-box attacks with full access to the model on five Whisper-family models (Radford et al., 2023), including Whisper-tiny, Whisper-base, Whisper-small, Whisper-medium, and Whisper-large, all obtained from HuggingFace. To benchmark the proposed MORE approach, we compare it against five strong white-box attack baselines: PGD (Olivier & Raj, 2022b), MI-FGSM (Dong et al., 2018), VMI-FGSM (Wang & He, 2021), the speech-aware gradient optimization (SAGO) method (Gao et al., 2024), and SlothSpeech (Haque et al., 2023).

**Experimental Setup.** In MORE, the hyperparameters $I$ and $K_a$ are set to 10 and 50, respectively. To ensure imperceptibility to human listeners, we set the perturbation magnitudes $\epsilon$ to 0.002 and 0.0035, which correspond to average signal-to-noise ratios (SNRs) of 35 dB and 30 dB, respectively, both within the range generally considered inaudible to humans (Gao et al., 2024). All experiments are conducted using a NVIDIA H100 GPU.

**Evaluation Metrics.** To evaluate ASR accuracy degradation, we adopt word error rate (WER) as the metric, which quantifies the proportion of insertions, substitutions, and deletions relative to the number of ground-truth words. Given that adversarial transcriptions may be excessively long, we truncate the predicted sequence to match the length of the reference text to better evaluate accuracy degradation in the initial portion of the output, where meaningful recognition should occur; WER values exceeding 100.00% are capped at 100.00% for normalization. Higher WER indicates lower ASR performance and thus a more effective accuracy attack. Efficiency attack performance is measured by the length of the predicted text tokens, where a greater length indicates a more effective efficiency attack.

## 5 RESULTS AND DISCUSSION

We evaluate MORE against state-of-the-art baselines on both accuracy and efficiency across multiple ASR models, examine its robustness under different SNR levels, and conduct ablations to analyze component contributions. A case study with adversarial examples and decoded transcriptions further illustrates its effectiveness.

| Attack methods | WER | Length | WER | Length |
|---|---|---|---|---|
| **SNR** | 35 | 35 | 30 | 30 |
| **MORE** | 90.85 | **296.66** | 94.80 | **326.62** |
| MORE - $\mathcal{L}_{\text{acc}}$ | 27.58 | 293.03 | 34.33 | 296.91 |
| MORE - $\mathcal{L}_{\text{eff}}$ | 93.63 | 30.60 | 96.56 | 31.79 |
| MORE - $\mathcal{L}_{\text{EOS}}$ | 93.92 | 233.84 | **96.68** | 269.36 |
| MORE - $\mathcal{L}_{\text{REDO}}$ | 92.42 | 120.67 | 95.72 | 146.54 |
| MORE - $\mathcal{L}_{\text{REDO}}$ - $\mathcal{L}_{\text{acc}}$ | 47.10 | 116.05 | 61.62 | 123.58 |
| MORE - $\mathcal{L}_{\text{EOS}}$ - $\mathcal{L}_{\text{acc}}$ | 6.97 | 270.21 | 7.96 | 307.38 |
| Clean | 5.34 | 18.55 | 5.34 | 18.55 |

Table 3: Ablation study of the proposed MORE approach by removing different components, evaluating adversarial attack performance in terms of accuracy (WER) and efficiency (length) on the LJ-Speech dataset under SNR of 35 dB and 30 dB. The symbol '−' denotes component removal.

## 5.1 Main Results Across Different Attacks and Models

We compare MORE with SOTA baselines across multiple ASR models, with results on two datasets at 30 dB and 35 dB shown in Table 1 and Table 2.

Our MORE approach consistently achieves superior efficiency attack performance—generating significantly longer transcriptions while maintaining high WER for accuracy degradation across both SNR levels. Specifically, accuracy-oriented baselines (PGD, SAGO, MI-FGSM, and VMI-FGSM) achieve substantially lower transcription lengths (e.g., 31.65 vs. our 300.13), highlighting the effectiveness of our novel repetitive encouragement doubling objective (REDO). Compared to Sloth-Speech, a baseline specifically designed for efficiency attacks, MORE still achieves significantly longer outputs (e.g., 208.52 vs. 65.85), underscoring the efficacy of our doubling loss design in REDO to effectively induce repetitive and extended transcriptions.

Notably, on the robust Whisper-large model, our MORE approach exhibits exceptional performance in both accuracy and efficiency dimensions. It achieves higher accuracy degradation (WER of 53.72 compared to about 30 for accuracy-oriented attack baselines) while producing average transcription lengths of 301.47—roughly 10 times longer than accuracy-oriented baselines and approximately 3.8 times longer than SlothSpeech (79.78). Additionally, SlothSpeech's weaker accuracy degradation (WER of 46.80 vs. MORE's 91.01) highlights the limitations of optimizing solely for efficiency and underscores the necessity of incorporating accuracy objectives for comprehensive attacks. These findings validate the robustness and utility of our proposed multi-objective optimization approach.

## 5.2 Impact of SNR Conditions

We further investigate the effect of varying SNR levels on attack effectiveness. By comparing results under 30 dB SNR (Table 2) and 35 dB SNR (Table 1), we observe that attacks under 30 dB SNR generally yield stronger performance, characterized by higher WER and significantly longer transcriptions (length). This confirms that lower SNR (i.e., noisier conditions) provides a more favorable environment for adversarial perturbations to succeed.

## 5.3 Ablation Study

We conduct an ablation study to assess the contribution of each component in the proposed MORE approach (Table 3). Eliminating the accuracy attack loss leads to the collapse of the accuracy attack performance (from 90.85 to 27.58), whereas the efficiency attack remains effective. This indicates that the proposed REDO and EOS objectives are critical for sustaining the efficiency attack. Removing the $\mathcal{L}_{\text{EOS}}$ leads to a decline in efficiency performance from 296.66 to 233.84, indicating that the EOS loss facilitates longer output generation by the Whisper model. However, both accuracy (WER > 90) and efficiency (length > 200) attacks remain effective, suggesting that REDO and multi-objective optimization (MO) with accuracy objective are more critical than EOS loss. When REDO is removed, efficiency performance drops drastically from 296.66 to 120.67, highlighting REDO's

| Attacks | Adversarial decoded text transcriptions from adversarial speech | Length | WER |
|---|---|---|---|
| Clean | The Roman type of all these printers is similar in character. | 12 | 0.00 |
| PGD | of Rutland who tried before gallopy was ventenies for contemporary priorath probes. | 18 | 100.00 |
| SlothSpeech | The Roman type of all these. The Roman type of all these. We're still in a learning career. We're still in a learning career. We're still in a learning career. We're still in a learning career. | 42 | 72.73 |
| SAGO | the Rolling Tide. For more on our latest ventilators for summary of Fire Esther. | 17 | 90.91 |
| VMI-FGSM | Roman plight, although he's fengomed for the century of my affairs. | 14 | 100.00 |
| MI-FGSM | and Rutland tied for Broadway's ventilence for Senior Order of Fire efforts. | 18 | 100.00 |
| MORE | her mind, her voice, her voice, and her voice, and her voice, and her voice, and her voice, and her voice, and her voice, and her voice, .......**(repeat 100 times) and her voice**, ...... and her voice, and her voice, and her voice, and | **334** | **100.00** |

Table 4: Comparison of decoded transcriptions from adversarially generated speech samples across different attacks: clean, PGD, SlothSpeech, SAGO, VMI-FGSM, MI-FGSM and the proposed MORE approaches. Clean represents the original text of the clean speech sample.

central role in promoting structured repetition. Meanwhile, the WER remains above 90, suggesting that sacrificing a small amount of accuracy can substantially benefit efficiency—reflecting the inherent trade-off in balancing the two objectives. Eliminating $\mathcal{L}_{\mathrm{acc}}$ in addition to $\mathcal{L}_{\mathrm{REDO}}$ sharply degrades both accuracy attack (WER 90.85 → 47.10) and efficiency (length 296.66 → 116.05), underscoring the importance of the proposed MO in jointly supporting both attack objectives. Removing both accuracy loss and EOS leads to a WER of 6.97, indicating complete failure of accuracy attacks. However, efficiency remains high (270.21), showing that REDO alone can still sustain efficiency attacks without MO with accuracy loss or EOS. Removing all efficiency attack components—EOS and REDO—causes efficiency to fail completely (length drops to 30.60), with only the accuracy attack (WER 93.63) remaining effective. This confirms that all efficiency design components are essential for achieving successful efficiency degradation. Overall, all components are critical and effectively contribute to the success of the attacks against the victim ASR models.

### 5.4 CASE STUDY: ADVERSARIAL SAMPLES

To qualitatively demonstrate the effectiveness of MORE, we present decoded adversarial transcriptions in Table 4. Unlike other baselines, which produce either incorrect outputs, MORE generates a fully incorrect transcription with a structured repetition of the sentence *"and her voice"* over 100 times, resulting in a length of 334 and a WER of 100.00. This showcases the strength of our proposed multi-objective optimization and the repetitive doubling encouragement objective in simultaneously disrupting transcription accuracy and inducing extreme inefficiency through systematic and semantically coherent redundancy. More case studies can be found in Appendix A.

## 6 CONCLUSION

We propose **MORE**, a novel adversarial attack approach that introduces multi-objective repulsion-anchoring optimization to hierarchically target recognition accuracy and inference efficiency in ASR models. MORE integrates a periodically updated repetitive encouragement doubling objective (REDO) with end-of-sentence suppression to induce structured repetition and generate substantially longer transcriptions while retaining effectiveness in accuracy attacks. Experimental results demonstrate that MORE outperforms existing baselines in efficiency attacks while maintaining comparable performance in accuracy degradation, effectively revealing dual vulnerabilities in ASR models. The code will be made publicly available upon acceptance.

## ETHICS STATEMENT

All authors have read and agree to adhere to the ICLR Code of Ethics. We understand that the Code applies to all conference participation, including submission, reviewing, and discussion. This work does not involve human-subject studies, user experiments, or the collection of new personally identifiable information. All evaluations use publicly available research datasets under their respective licenses. No attempts were made to attack deployed systems, bypass access controls, or interact with real users.

Our contribution, **MORE**, is an adversarial method that can degrade both recognition accuracy and inference efficiency of ASR systems. While our goal is to advance robustness research and stress-test modern ASR models, the same techniques maybe misused to (i) impair assistive technologies (e.g., captioning for accessibility), (ii) disrupt safety- or time-critical applications (e.g., clinical dictation, emergency call transcription, navigation), or (iii) increase computational costs for shared services via artificially elongated outputs. We do not provide instructions or artifacts intended to target any specific deployed product or service, and we caution that adversarial perturbations, especially those designed to be inconspicuous, present real risks if applied maliciously.

To reduce misuse risk and support defenders, we suggest that some concrete defenses should be integrated into ASR systems: decoding-time safeguards such as repetition/loop detectors; input-time defenses such as band-limiting; and training-time strategies such as adversarial training focused on EOS/repetition pathologies. We will explore some targeting defense mechanisms against **MORE** in future work.

## REPRODUCIBILITY STATEMENT

We take several steps to support independent reproduction of our results. Algorithmic details for **MORE** are provided in Sec. 3 and further clarified in the Appendix B. Dataset choices and preprocessing (LibriSpeech & LJ-Speech, first 500 utterances per set, 16 kHz resampling) are specified in Sec. 4, while exact model variants (Whisper-tiny/base/small/medium/large from HuggingFace), hardware, perturbation budgets/SNRs, and all attack hyperparameters are detailed in Sec. 4. The evaluation protocol is defined in Sec. 4. We will not include the code archive in the submission due to proprietary requirements. Upon acceptance, we will release a public repository mirroring the anonymous package as soon as we get permission.

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

## A   MORE ADVERSARIAL TRANSCRIPTIONS

To showcase the effectiveness of MORE, we present more decoded adversarial transcriptions in Table 5 and Table 6.

| Sample 1 | Adversarial decoded text transcriptions from adversarial speech | Length | WER |
|---|---|---|---|
| Clean | in fourteen sixty-five sweynheim and pannartz began printing in the monastery of subiaco near rome | 22 | 0.00 |
| PGD | With a 14-a-tip to the dog, framed by its and clan herbs begin in renting a Ramona scurrying, home-or-dity-actyl-healed blowing. | 31 | 100 |
| SlothSpeech | In 1365. Twain, and patterns. We began printing in the monastery on Superacro. We are going to be here. We are going to be here. We are going to be here. We are going to be here. We are going to be here. We are going to be here. We are going to be here. | 32 | 57.14 |
| SAGO | In recaing 50 5, engraved ideas and claim herbs begin renting a maimanda scurrant you'll be able to keep the app's raw huge role. | 32 | 92.86 |
| VMI-FGSM | The importance in tips to guide, 3D appliance and clan-obes video granting a new monoscaring different types of CDR tools here in the life. | 30 | 100 |
| MI-FGSM | With the important tip to divide, engraved piles and clan herbs begin grinsling a Murmana scurin per Cynthia of Hugh Bully. | 29 | 100 |
| MORE | Say 14-2 to the yard. So, I am glad herbs begin in the printing, they were modest, they're in the home, I just keep the actual yield going. Well, I'm glad. Well, I'm glad. Well, I'm glad. Well, I'm glad. Well, I'm glad. Well, I'm glad. Well, I'm glad. Well, I'm glad. Well, I'm glad. ...**(repeat 50 times) Well, I'm glad.** ... Well, I'm glad. Well, I'm glad. Well, I'm glad. Well, I'm glad. Well, I'm glad. Well, I'm glad. Well, I'm glad. Well, I'm glad. Well, I'm glad. Well | **303** | **100.00** |

| Sample 2 | Adversarial decoded text transcriptions from adversarial speech | Length | WER |
|---|---|---|---|
| Clean | they printed very few books in this type, three only; but in their very first books in rome, beginning with the year fourteen sixty-eight | 26 | 0.00 |
| PGD | There's quite a very few dobs on the street. It's very close to New York, so I'm more alanists. If you're in the Euro-Saushen, I'm so creative. | 39 | 100.00 |
| SlothSpeech | They printed very pure robsimbus type. Three only. And they're very first robs in Rome. You're getting with the year 14, 16-8. | 31 | 52.17 |
| SAGO | I print a very few dobs on the scyphs. But it's in there there tortiros you can morelice, within any effort near the coat your rooms might create. | 37 | 52.17 |
| VMI-FGSM | And as front have there, with your bumps and best drives, the case also may fax them their source-of-fit off some well-ince if the payment Okuniro 40 or a safety aid. | 37 | 100.00 |
| MI-FGSM | There's currently as very few dobs on the street flaps in their very torti-doxing Berliners, depending of a year of caffeine I'm so pleased. | 33 | 86.96 |
| MORE | There's quite a very few dubs on the street. It's a great place. You seem very close to York, so I'm more alanists. It's the band-in-the-in-a-day. Are you or I? So, I'm 16, I'm 6, I create. It's a great place. It's a great place. It's a great place. It's a great place. ...**(repeat 55 times) It's a great place.** ... It's a great place. It's a great place. It's a great place. It's a great place. It's a great place. It's a great place. It's | **369** | **100.00** |

Table 5: Comparison of decoded transcriptions from adversarially generated speech samples across different attacks: clean, PGD, SlothSpeech, SAGO, VMI-FGSM, MI-FGSM and the proposed MORE approaches. Clean represents the original text of the clean speech sample.

| Sample 3 | Adversarial decoded text transcriptions from adversarial speech | Length | WER |
|---|---|---|---|
| Clean | and it was a matter of course that in the middle ages, when the craftsmen took care that beautiful form should always be a part of their productions whatever they were | 32 | 0.00 |
| PGD | If Anif wasn't battle with court, he's cut, can him believe dunesque? By McClack to slimped up K-9's ex to use with little pillows to be beldled his beat if lag and bail for gothician in his Trump's Lloyd-Avergen alone. | 60 | 96.77 |
| SlothSpeech | And it was a matter of course, in the middle ages, when the craftsmen took care that beautiful forms should rollers be a part of their productions, whatever they are. But everything. Yeah, you're not ..**(repeat 17 times) Yeah, you're not.** ... Yeah, you're not. Yeah, you're not. | 114 | 16.13 |
| SAGO | And until some battle with the Polish scut Hindu-dulisciensky by McPlac slimed some care and ex-eutectful of toarungs, be thrilled he's be a flag of bamper gothish in his tomb, to play out a dang horror. | 60 | 93.55 |
| VMI-FGSM | If anything is in battle with court he's gun kingdom of the Alliance Samuel Pratt has limped up Keanu and read youth with reform on each of the children who has beat up flag on boyfriend on to shamed us and point out your don't you lotta? | 55 | 100 |
| MI-FGSM | I don't know if hasn't battled with the poor, Scott, given with the diligence. By McClagg slimps, he hears, ends up doing to a toiling, shrid the soldier, he's beat up fog and bounced her goth or shin his stance and deploy everything, hoda.. | | |
| MORE | I don't know if I was in battle with court. He's got... He's got him in the lead, do you? I'm a clapped, it's slim, it's okay, I'm exactly... He's like, it's so own, it's a bit older, he's a bit flack, it's a bit off the gothician, it's strong. What else is this? ... **(repeat 66 times) What else is this?** ... What else is this is this? What else is this is this is this is this is this is this is this is this | **363** | **100.00** |

| Sample 4 | Adversarial decoded text transcriptions from adversarial speech | Length | WER |
|---|---|---|---|
| Clean | and which developed more completely and satisfactorily on the side of the "lower-case" than the capital letters. | 20 | 0.00 |
| PGD | I've entered Richard of Omen with her work from 2018 and status factor audio. My style Roman roman rumpims Dominic Pappett, all to my idealist. | 36 | 100.00 |
| SlothSpeech | And which denote more completely and satisfactorily, oh, beside the molecules? You've done the capital. .......**(repeat 8 times) You've done the capital.** ...... You've done the capital. | 63 | 61.11 |
| SAGO | and which the Omen of the Fomor from P-plea Mensec satisfactory are postiad Roman root pins. No more powerful of her influence. | 31 | 88.89 |
| VMI-FGSM | If you're a richer dog old, for more can keep these and to have a spare theory, I'm not today Roman Oruk winds. Roman, we're glad to be here. | 35 | 100.00 |
| MI-FGSM | You've had a rich devolt of her warmth from cleaved deno de téras diasporoia, her stayered Roman rificims. No meekly out the gludermervialis. | 43 | 100.00 |
| MORE | I've had a witcher development with her work on 22 and excited as faculty earlier. My style at Roman Roman pimps. Gonna make it happen. I'm all you're glad you're leaving. I'm glad you're leaving. ...**(repeat 54 times) I'm glad you're leaving.** ... I'm glad you're leaving. I'm glad you're | **387** | **100.00** |

Table 6: Comparison of decoded transcriptions from adversarially generated speech samples across different attacks: clean, PGD, SlothSpeech, SAGO, VMI-FGSM, MI-FGSM and the proposed MORE approaches. Clean represents the original text of the clean speech sample.

## B COMPLEXITY ANALYSIS

**Scope and assumptions.** We analyze two costs: (i) *attack-time* compute to craft $\delta$ via Algorithm 1, and (ii) *victim-time* compute when the ASR model decodes on $X+\delta$. Our derivation accounts for encoder self-attention, decoder self-/cross-attention, feed-forward layers, vocabulary projection/softmax, greedy decoding used to materialize doubled targets, and the backward/forward constant factor. We express totals in terms of model hyperparameters and the scheduling parameters $(K, K_a, D)$ defined in Algorithm 1, and we refer to the objectives in Eqs. 5, 6, 8, and 9.

**Notation.** Let $F$ be the number of encoder time steps (e.g., log-Mel frames). Encoder depth/width/FF width/heads are $N_e, d_e, d_{\text{ff},e}, h_e$; decoder counterparts are $N_d, d_d, d_{\text{ff},d}, h_d$. Vocabulary size is $V$. The total PGD steps are $K$, Stage 1 steps are $K_a$, and the Stage 2 block length is $D$ (Algorithm 1). Stage 2 has $M = \lceil (K-K_a)/D \rceil$ blocks indexed by $m = 1, \ldots, M$. We denote by $B_m$ the base segment (non-EOS) used to build the doubled target in block $m$, and by $L_m = |B_m|$ its length. The doubly repeated target in block $m$ has length $\bar{\ell}_m = 2L_m$. We write $\kappa \in [2, 3]$ (empirically range) for the backward/forward multiplier. Let $T$ be the number of raw samples in $X$ (used only for the $O(T)$ PGD update).

**On the construction of doubled targets.** Eq. 8 defines $\bar{Y}_i$ via $\hat{y}_{\lfloor \frac{i}{D} \rfloor}$ but does not specify how the hypothesis $\hat{Y}$ is obtained at the start of each block. To make Eq. 8 operational, we explicitly realize $\hat{Y}$ with a *greedy decode* on the current perturbed input:

$$\hat{Y}^{(m)} = \text{GREEDYDECODE}(f, X+\delta), \tag{10}$$

$$B_m = \text{STRIPEOS}(\hat{Y}^{(m)}), \tag{11}$$

$$\bar{Y}^{(m)} = B_m \parallel B_m, \tag{12}$$

$$\bar{\ell}_m = 2|B_m|. \tag{13}$$

Here, Eq. 10 computes decoding the perturbed input $X+\delta$ with the victim model $f(\cdot)$ using greedy decoding (selecting the most probable token at each step until termination). In Eq. 11, $\text{STRIPEOS}(\cdot)$ removes the terminal EOS token from the decoded hypothesis, leaving only the content-bearing tokens. The notation "$\parallel$" denotes sequence concatenation, so Eq. 12 constructs the doubled sequence by repeating $B_m$ back-to-back. Finally, $\bar{\ell}$ calculated by Eq. 13 is the length of this doubled target. This makes Eq. 8 explicit and adds a per-block greedy-decoding cost accounted for below.

**Per-pass building blocks.** We use standard Transformer accounting; QKV and output projections are absorbed in big-$\mathcal{O}$ terms. For a decoder sequence length $\ell$,

$$C_{\text{enc}}(F) = \mathcal{O}\Big( N_e\big( F^2 d_e + F d_e d_{\text{ff},e} \big) \Big), \tag{14}$$

$$C_{\text{dec-TF}}(\ell, F) = \mathcal{O}\Big( N_d\big( \ell^2 d_d + \ell F d_d + \ell d_d d_{\text{ff},d} \big) \Big), \tag{15}$$

$$C_{\text{dec-gen}}(\ell, F) = \mathcal{O}\Big( N_d\big( \ell^2 d_d + \ell F d_d + \ell d_d d_{\text{ff},d} \big) \Big). \tag{16}$$

The encoder cost in Eq. 14, $C_{\text{enc}}(F)$, measures the cost of processing $F$ input frames, scaling as $F^2 d_e$ for self-attention and $F d_e d_{\text{ff},e}$ for feed-forward layers over $N_e$ encoder layers. The decoder cost under teacher forcing in Eq. 15, $C_{\text{dec-TF}}(\ell, F)$, captures the cost of processing a target sequence of length $\ell$, scaling quadratically in $\ell$ from decoder self-attention, linearly in $\ell F$ from cross-attention with encoder outputs, and linearly in $\ell d_{\text{ff},d}$ from feed-forward layers over $N_d$ decoder layers. The decoder cost under generation in Eq. 16, $C_{\text{dec-gen}}(\ell, F)$, has the same asymptotic form as teacher forcing since autoregressive decoding still requires self-attention, cross-attention, and feed-forward passes, though key–value caching can reduce constants in practice. In both modes, an additional $\mathcal{O}(\ell V)$ term arises from vocabulary projection and softmax, which is significant for large vocabularies but dominated by $\ell^2$ self-attention when $\ell$ is large. Both Eq. 15 and Eq. 16 scale as $\ell^2$ (self-attention) and as $\ell F$ (cross-attention). The vocabulary projection/softmax adds $\mathcal{O}(\ell V)$ per forward/backward; we keep it explicit when informative.

**Loss conditioning and reuse of passes.** Eqs. 5 and 9 are cross-entropy (CE) objectives and must be computed under *teacher forcing* to provide stable gradients. We assume CE terms use teacher forcing throughout. For Eq. 6, we define $P_L^v$ at the *last teacher-forced position* (so $L = \bar{\ell}_m$ in Stage 2). With Algorithm 1 summing $\mathcal{L}_{\text{EOS}} + \mathcal{L}_{\text{REDO}}$ each Stage 2 step, both losses share *one* forward/backward pass at length $\bar{\ell}_m$; no additional pass is required for EOS.

ATTACK-TIME COMPLEXITY

**Stage 1 (Accuracy; $K_a$ steps).** Each step backpropagates $\mathcal{L}_{\text{acc}}$ (Eq. 5) under teacher forcing on $Y$ of length $L_{\text{acc}} = |Y|$:

$$C_{\text{step}}^{(1)} = \kappa[C_{\text{enc}}(F) + C_{\text{dec-TF}}(L_{\text{acc}}, F)] + \mathcal{O}(L_{\text{acc}}V). \tag{17}$$

Optional early-stopping evaluations (e.g., greedy WER every $E$ steps) add

$$C_{\text{eval,S1}} \approx \left\lceil \frac{K_a}{E} \right\rceil \cdot \left( C_{\text{enc}}(F) + C_{\text{dec-gen}}(L_{\text{eval}}, F) + \mathcal{O}(L_{\text{eval}}V) \right), \tag{18}$$

where $L_{\text{eval}}$ is the decoded length at evaluation.

**Stage 2 (Efficiency; $K - K_a$ steps).** Stage 2 consists of $M$ blocks of $D$ steps. In each block $m$:

- *Greedy anchor (once per block).* Build $\bar{Y}^{(m)}$ by decoding $\hat{Y}^{(m)}$ and doubling (Eq. 8):

$$C_{\text{greedy},m}^{(2)} = C_{\text{enc}}(F) + C_{\text{dec-gen}}(L_m^{\text{gen}}, F) + \mathcal{O}(L_m^{\text{gen}}V), \quad \text{with } L_m^{\text{gen}} \approx L_m. \tag{19}$$

- *PGD steps (every step in the block).* Algorithm 1 uses the *sum* $\mathcal{L}_{\text{EOS}} + \mathcal{L}_{\text{REDO}}$ each step, with teacher-forced length $\bar{\ell}_m = 2L_m$. Hence, per step:

$$C_{\text{step}}^{(2)}(m) = \kappa[C_{\text{enc}}(F) + C_{\text{dec-TF}}(\bar{\ell}_m, F)] + \mathcal{O}(\bar{\ell}_m V), \tag{20}$$

and over $D$ steps:

$$C_{\text{block}}^{(2)}(m) = D\, C_{\text{step}}^{(2)}(m) = D\, \kappa[C_{\text{enc}}(F) + C_{\text{dec-TF}}(2L_m, F)] + \mathcal{O}(D\, 2L_m V). \tag{21}$$

Summing over blocks,

$$C_{\text{Stage2}} = \sum_{m=1}^{M} \left( C_{\text{greedy},m}^{(2)} + C_{\text{block}}^{(2)}(m) \right)$$

$$= \underbrace{\kappa(K - K_a)\, C_{\text{enc}}(F)}_{\text{encoder re-run each PGD step}} + \kappa N_d \sum_{m=1}^{M} \left[ D\,(2L_m)^2 d_d + D\,(2L_m)\,(Fd_d + d_d d_{\text{ff},d}) \right]$$

$$+ \sum_{m=1}^{M} C_{\text{greedy},m}^{(2)} + \mathcal{O}\!\left( V \sum_{m=1}^{M} D\,(2L_m + L_m^{\text{gen}}) \right). \tag{22}$$

**Growth envelopes for $L_m$.** The doubled-target curriculum encourages $L_m$ to increase across blocks.

- *Geometric (until cap).* In this case, $L_m$ grows by doubling until it reaches the cap $L_{\max}$:

$$L_m = \min\{L_0\, 2^{m-1},\ L_{\max}\}, \tag{23}$$

$$M^{\star} = \min\left\{ M,\ 1 + \lfloor \log_2(L_{\max}/L_0) \rfloor \right\}. \tag{24}$$

Then the sums are

$$\sum_{m=1}^{M} L_m = L_0(2^{M^{\star}} - 1) + (M - M^{\star})L_{\max}, \tag{25}$$

$$\sum_{m=1}^{M} L_m^2 = \frac{L_0^2}{3}(4^{M^{\star}} - 1) + (M - M^{\star})L_{\max}^2. \tag{26}$$

Plugging into Eq. 22, the self-attention term scales as $\Theta\!\left( D \cdot 4^{M^{\star}} \right)$ before saturation at $L_{\max}$.

- *Linear (until cap).* If $L_m = L_0 + (m-1)\Delta$ (capped at $L_{\max}$), then the uncapped sums are

$$\sum_{m=1}^{M} L_m = \tfrac{M}{2}\big(2L_0 + (M-1)\Delta\big), \tag{27}$$

$$\sum_{m=1}^{M} L_m^2 = ML_0^2 + L_0\Delta M(M-1) + \tfrac{\Delta^2}{6}(M-1)M(2M-1). \tag{28}$$

Here the dominant term in the self-attention sum is $\Theta(D\,M^3\Delta^2)$ when $\Delta > 0$.

**Total attack-time cost.** Combining stages,

$$C_{\text{attack}} = \kappa K_a[C_{\text{enc}}(F) + C_{\text{dec-TF}}(L_{\text{acc}}, F)] + C_{\text{eval,S1}} + C_{\text{Stage2}}, \tag{29}$$

with $C_{\text{Stage2}}$ calculated from Eq. 22. The PGD update is $\mathcal{O}(T)$ and negligible.

VICTIM-TIME (INFERENCE) COMPLEXITY

When decoding on $X+\delta$ without backprop, expected per-example compute is

$$C_{\text{infer}}(\ell_{\text{adv}}) = C_{\text{enc}}(F) + C_{\text{dec-gen}}(\ell_{\text{adv}}, F) + \mathcal{O}(\ell_{\text{adv}}V), \tag{30}$$

where $\ell_{\text{adv}}$ is the induced output length under the chosen decoding policy (greedy by default). Focusing on decoder-dominant terms, the slowdown relative to the clean case ($\ell_{\text{clean}}$) is

$$\frac{C_{\text{infer}}(\ell_{\text{adv}})}{C_{\text{infer}}(\ell_{\text{clean}})} \approx \frac{\ell_{\text{adv}}^2 + \ell_{\text{adv}}F + \ell_{\text{adv}}\frac{d_{\text{ff},d}}{d_d}}{\ell_{\text{clean}}^2 + \ell_{\text{clean}}F + \ell_{\text{clean}}\frac{d_{\text{ff},d}}{d_d}},$$

highlighting quadratic sensitivity to $\ell_{\text{adv}}$ from decoder self-attention. Under the geometric envelope above, $\ell_{\text{adv}}$ can grow proportionally to $2^{M^\star}L_0$ until capped by the implementation's maximum length, potentially yielding a $\Theta(4^{M^\star})$ increase in decoder FLOPs before saturation.

**Summary bound.** With $M = \lceil (K-K_a)/D \rceil$ and $\bar{\ell}_m = 2L_m$, the attack-time compute admits

$$C_{\text{attack}} \in \mathcal{O}\Big( \kappa K\, C_{\text{enc}}(F) + \kappa N_d\, D \sum_{m=1}^{M} \big( \underbrace{\bar{\ell}_m^2}_{4L_m^2} d_d + \bar{\ell}_m (Fd_d + d_d d_{\text{ff},d}) \big)$$

$$+ \sum_{m=1}^{M} C_{\text{greedy},m}^{(2)} + V \cdot D \sum_{m=1}^{M} \bar{\ell}_m \Big). \tag{31}$$

This bound separates: (i) repeated encoder passes linear in $K$ and $F$; (ii) decoder self-attention terms growing with $\sum_m L_m^2$ (the principal driver under elongation); and (iii) vocabulary and greedy-decoding overheads linear in sequence length and $V$.

# C  ADDITIONAL EFFICIENCY ANALYSIS OF MORE

To complement our analysis based on output length, we provide a more explicit characterization of the computational overhead induced by MORE in terms of floating-point operations (FLOPs).

## C.1  ESTIMATED INFERENCE FLOPS UNDER MORE VS. BASELINES

Following standard FLOPs estimates for Transformer models, i.e., approximately $2 \cdot N$ FLOPs per generated token for a model with $N$ parameters (Casson), we approximate the per-example inference cost (encoder + decoder) for Whisper. Since the encoder runs once per utterance while the decoder runs once per output token, the relative increase in FLOPs is dominated by the increase in output length caused by MORE. In typical (non-attack) conditions on LibriSpeech, Whisper produces transcriptions roughly the same length as the reference transcript—about 22 tokens on average per utterance. Based on Table 1, MORE can induce $10\times$ to $14\times$ longer transcripts compared to normal

Table 7: Estimated per-example inference FLOPs (in billions) for Whisper under baseline decoding vs. MORE on LibriSpeech, using standard Transformer FLOPs estimates [1].

| Model | Params (M) | Baseline tokens (avg) | MORE tokens (avg) | Baseline FLOPs (G) | MORE FLOPs (G) | × Increase |
|---|---|---|---|---|---|---|
| Tiny | 39 | 22 | 296 | 1.7 | 23.1 | 13.5× |
| Base | 74 | 22 | 300 | 3.3 | 44.4 | 13.6× |
| Small | 244 | 22 | 214 | 10.7 | 104.4 | 9.7× |
| Medium | 769 | 22 | 234 | 33.8 | 359.9 | 10.6× |
| Large | 1550 | 22 | 301 | 68.2 | 933.1 | 13.7× |

outputs across different Whisper model sizes. Using the parameter sizes of Whisper models and the average output lengths observed in our experiments, we obtain the following per-example FLOP on the LibriSpeech dataset.

Here we approximate (1) FLOPs per token $\approx 2 \cdot N_{\text{params}}$; (2) Total FLOPs per example $\approx \text{FLOPs}_{\text{encoder}} + (\#\text{tokens}) \cdot 2 \cdot N_{\text{params}}$. And we use the empirical baseline vs. MORE token lengths from our experiments. These estimates show that, across all Whisper sizes, MORE increases per-example inference compute by roughly an order of magnitude ($\approx 9\text{--}14\times$), purely by forcing the model to generate much longer, repetitive transcripts. This quantifies an *efficiency vulnerability*: MORE does not just degrade accuracy, but also inflates the FLOPs required for inference, threatening the real-time and resource efficiency of ASR deployments.

## C.2 ANALYSIS OF ASR INFERENCE TIME

In Fig. 2, we profile the inference time of the Whisper-Large model as a function of the number of output tokens. The inference time increases almost linearly with the output length. Moreover, Whisper-Large pads all inputs shorter than 30 seconds to a fixed 30-second window before processing, so utterances shorter than 30 seconds incur identical encoding time, making the output length the only factor that determines the overall resource consumption. These observations support our motivation to maximize the output length in order to induce the greatest possible waste of computational resources.

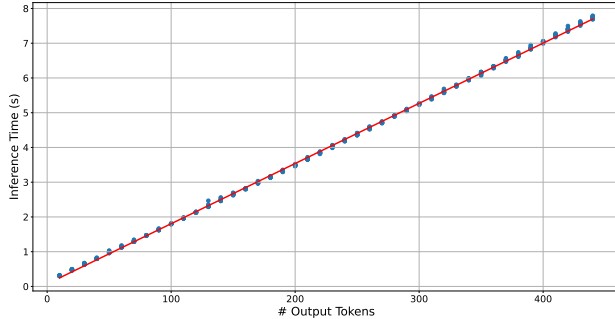

Figure 2: Inference Time of Whisper-large versus output length.

## D THE USE OF LARGE LANGUAGE MODELS

We used a large language model (ChatGPT) solely as a writing assist tool for grammar checking, wording consistency, and style polishing of author-written text. All technical content, results, and conclusions originate from the authors. Suggested edits were reviewed by the authors for accuracy and appropriateness before inclusion. No confidential or proprietary data beyond manuscript text was provided to the tool. This disclosure is made in accordance with the venue's policy on LLM usage.

