# OpenReview forum: "MORE: Multi-Objective Adversarial Attacks on Speech Recognition"
_ICLR.cc/2026/Conference — Submitted to ICLR 2026_

### Official Review · Reviewer_v3bB · 2025-10-26

**Soundness:** 3
**Presentation:** 3
**Contribution:** 3
**Rating:** 4
**Confidence:** 4

**Summary:**

This paper proposes MORE (Multi-Objective Repetitive Doubling Encouragement), an adversarial attack designed to jointly degrade accuracy and efficiency of automatic speech recognition (ASR) systems, particularly the Whisper family. The authors introduce a two-stage hierarchical optimization framework: (1) a “repulsion” stage that maximizes transcription errors using cross-entropy loss, and (2) an “anchoring” stage that prolongs decoding by suppressing the end-of-sentence (EOS) token and introducing a Repetitive Encouragement Doubling Objective (REDO) that induces periodic repetition in the generated transcripts.

**Strengths:**

The paper explores a less-examined dimension of ASR robustness: efficiency robustness, i.e., how adversarial perturbations can cause excessive computation via unending decoding loops.

The multi-objective formulation combining accuracy degradation and inference slowdown is conceptually interesting and fills a niche left by prior single-objective adversarial attacks such as PGD, MI-FGSM, and SlothSpeech.

Introducing a hierarchical optimization (repulsion–anchoring) and a specialized REDO mechanism to induce repetition is a creative way to operationalize efficiency degradation.

The proposed algorithm is clearly described with mathematical formulation (Eqs. 5–9, Algorithm 1) and supported by a thorough ablation (Table 3) showing the effect of removing each component.

Comparative results on five Whisper model sizes (tiny→large) and two datasets show consistent trends: MORE produces much longer transcripts while keeping WER extremely high.

**Weaknesses:**

Only whisper family is tested.
All experiments are confined to the Whisper family, which share the same encoder–decoder architecture and CTC-free autoregressive decoding. This limits the claim of “comprehensive ASR robustness analysis.” Evaluating at least one non-Whisper or CTC-based model (e.g., wav2vec 2.0 + CTC, DeepSpeech-CTC) would strengthen generality.

The REDO repetition objective relies on autoregressive token prediction and EOS suppression.
For ASR systems using sliding-window inputs or CTC loss, token duplication beyond the receptive window would not be guaranteed, since predictions are frame-aligned and not dependent on previous outputs.
The paper does not explain how the method would ensure repetition or efficiency degradation under such architectures.

The paper claims to be the first “multi-objective attack,” but the proposed combination of accuracy and efficiency optimization largely merges existing adversarial attack objectives (for WER maximization) with repetition-inducing or EOS-suppression tricks that have been studied in text generation (e.g., Xu et al., 2022).
The paper does not articulate what fundamentally distinguishes this setup from prior adversarial or repetition-loop attacks.
The “hierarchical repulsion–anchoring” design could be presented as a general training heuristic rather than a conceptual breakthrough.

Nits: Break long sentence like " To enhance attack effectiveness, we not only reduce the likelihood of EOS but also explicitly emphasize a competing token, the token with the second largest probability, since reinforcing this alternative both reduces EOS dominance and steers the model toward an alternative continuation."

**Questions:**

Scope and unique challenge:
What makes attacking efficiency fundamentally different or more challenging than simply maximizing output length or suppressing EOS?
How does the proposed hierarchical optimization overcome specific difficulties not addressed by existing repetition-loop analyses?

Model generality:
Why were only Whisper models tested? Have you attempted the same attack on CTC-based ASR systems (e.g., wav2vec 2.0 CTC, Conformer)?
Would REDO or EOS-suppression work when the model’s decoding process lacks autoregression?

Repetition mechanism validity:
For models using a sliding-window decoder and limited spectrogram context, how can you ensure that repeated content remains predictable given input truncation? Does the attack rely on the model’s cache or hidden-state persistence? If so, how does window size affect repetition length?

Measurement of “efficiency degradation”:
Could you provide actual runtime or FLOP comparisons, not just token lengths, to substantiate efficiency loss?
How do hardware or decoding-beam settings influence the attack effect?


The paper cites SlothSpeech as the only efficiency-targeting method, but earlier “loop attacks” or “repetition failures” in text generation (e.g., Xu et al., 2022) already described similar failure modes. How is MORE conceptually distinct from combining an existing adversarial loss with an EOS penalty?

Potential defenses: Given the repetition-loop nature of the attack, have you tested whether simple decoding constraints (e.g., repetition penalty, token-frequency cap, or beam search with anti-loop heuristics) neutralize the effect?

---

> ### Author Response · Authors · 2025-11-21
> **Response to weakness (1) (2)**
>
> Thank you for this thoughtful comment. We focus on the Whisper family because it is one of the most widely deployed ASR architectures today, both as a standalone recognizer and as the speech front-end for many multimodal LLMs. Evaluating attacks across five Whisper variants of different scales already provides a meaningful view of how large-scale autoregressive ASR models behave under multi-objective attacks.
>
> We agree that extending the evaluation to non-Whisper architectures, such as CTC-based systems, would offer additional insights. However, such models rely on fundamentally different decoding mechanisms. CTC systems produce frame-aligned outputs and do not use autoregressive token prediction or EOS-based termination, meaning that REDO-driven repetition and EOS suppression would not apply in the same manner. Exploring how multi-objective efficiency attacks should be reformulated under CTC-style decoding therefore requires methodological changes that are outside the scope of this paper.
>
> We appreciate the reviewer’s suggestion and view the investigation of non-autoregressive and CTC-based ASR architectures as an important avenue for future work. Our present goal is to analyze and expose efficiency vulnerabilities in widely used autoregressive ASR models, and Whisper provides a representative and practically relevant testbed for this purpose.

---

> ### Author Response · Authors · 2025-11-21
> **Response to weakness (3) (4)**
>
> Weakness 3:
>
> Thank you for the insightful comment. We appreciate the opportunity to clarify the distinction between MORE and prior work, including repetition-loop analyses such as Xu et al. (2022). Xu et al. focus on understanding and mitigating naturally occurring repetition loops in text generation models. Their work neither involves adversarial perturbations nor targets ASR systems, and it does not address efficiency degradation or multi-objective adversarial settings. We cite their findings only as motivational evidence that autoregressive models exhibit repetition-amplifying dynamics that can be intentionally exploited by an adversary.
>
>
> In contrast, our objective is to construct adversarial perturbations on raw speech inputs that simultaneously manipulate two outcomes of an ASR model: transcription accuracy and inference efficiency. This requires actively steering the model into a specific failure mode (structured repetition and delayed termination) while ensuring the perturbation remains imperceptible. To our knowledge, prior ASR attacks have examined only accuracy degradation or only efficiency degradation (as in SlothSpeech), but not both jointly. No prior work has leveraged repetition-loop behavior in an adversarial context to induce controlled, large-scale sequence expansion in ASR decoding.
>
>
> The hierarchical repulsion–anchoring design is introduced not as a conceptual breakthrough in optimization but as a practical mechanism tailored to ASR attacks. It stabilizes the interaction between two objectives that operate on different gradient scales and different parts of the decoding process. Without this staged structure, the efficiency-oriented objective can dominate early updates, causing the model to remain too close to the ground-truth trajectory and reducing attack reliability. The repulsion–anchoring formulation therefore serves as an attack-specific strategy for reliably activating repetition-based vulnerabilities in autoregressive ASR models.
>
> Weakness 4:
>
> Thank you for the suggestion. We agree that the original sentence was overly long and could be made clearer. We have revised it into shorter, more readable sentences. The updated text is included below and highlighted in blue in the revised manuscript.
>
> **Revised version**:
>
> To enhance attack effectiveness, we reduce the likelihood of the EOS token. In addition, we increase the probability of the competing token with the second-largest likelihood. Reinforcing this alternative token not only diminishes EOS dominance but also guides the model toward continued generation.

---

> ### Author Response · Authors · 2025-11-21
> **Response to question (1)**
>
> Thank you for the insightful question. Attacking efficiency in ASR involves challenges that go beyond simply maximizing output length or suppressing the EOS token. SlothSpeech, which we include as a baseline, represents an EOS-only efficiency attack: it reduces the probability of EOS but does not provide any structured mechanism for guiding the continuation. As a result, EOS suppression alone often leads to unstable or low-confidence generation, and the model may quickly drift into short, incoherent continuations rather than producing the long, predictable expansions required for a strong efficiency attack.
>
> Our results in Tables 1, 2, and 4 validate this limitation: SlothSpeech consistently yields shorter expansions and weaker accuracy degradation compared with MORE. While SlothSpeech demonstrates that EOS suppression can extend decoding to some extent, it does not reliably induce large-scale repetition or consistent efficiency degradation across models.
>
> The key challenge is that efficiency gradients are extremely sparse and operate on a single termination token, while the autoregressive decoder strongly favors the correct next-token distribution. This makes efficiency attacks fundamentally difficult to optimize, especially under strict perturbation constraints on raw audio. MORE addresses this by combining the accuracy-degradation step with the REDO repetition objective. The repulsion stage first moves the model away from the correct decoding path, making it more susceptible to efficiency manipulation. The anchoring stage then uses REDO to enforce structured repetition, which leverages transformer repetition dynamics to produce stable, controlled, and significantly longer outputs.
>
> Existing repetition-loop analyses (e.g., Xu et al., 2022) investigate natural repetition behaviors in text generation, but they do not involve adversarial perturbations, audio-based constraints, or efficiency attacks. Our work is the first ASR attack study to examine and adversarially exploit repetition mechanisms for efficiency degradation.
>
> In summary, attacking efficiency requires more than EOS suppression or length maximization. MORE overcomes the inherent weaknesses of EOS-only methods through its hierarchical structure and controlled repetition mechanism, and this difference is empirically reflected in our strong performance compared with SlothSpeech across all evaluated models.

---

> ### Author Response · Authors · 2025-11-21
> **Response to question (2)**
>
> Thank you for your comments. We focus on Whisper because it is one of the most widely used ASR models today and represents a strong, large-scale autoregressive architecture. Our goal is to study multi-objective vulnerabilities in this practically relevant class of models rather than to evaluate every ASR paradigm. We did not experiment with CTC-based systems such as wav2vec 2.0. These models use different decoding mechanisms without autoregressive token prediction or EOS-based termination, so REDO and EOS suppression would not operate in the same way. Extending multi-objective efficiency attacks to CTC-style models would require a different formulation and is an interesting direction for future work.

---

> ### Author Response · Authors · 2025-11-21
> **Response to question (3)**
>
> Thank you for the question. Our study focuses on standard offline Whisper models, which perform full-context autoregressive decoding rather than sliding-window or streaming decoding. As a result, repetition in our setting does not depend on truncated spectrogram windows, cache resetting, or limited-context hidden states. The repeated content generated by REDO arises from the model’s full-sequence decoder context, and therefore does not rely on long-speech buffering or streaming mechanisms. Because we do not target long-form or streaming ASR architectures, we do not encounter the window-size constraints mentioned in the question. Extending multi-objective efficiency attacks to sliding-window or streaming ASR systems  would require a different design and is an interesting direction for future work.

---

> ### Author Response · Authors · 2025-11-21
> **Response to question (4)**
>
> Thank you for the question. We have included preliminary efficiency analyses to Appendix C. Specifically, we compare the estimated inference FLOPs of MORE and the baselines. We also measure the actual inference time of samples, which better reflects real-world computational efficiency, as shown in Figure 2 in Appendix C on page 18. Overall, we have the conclusion that runtime almost correlates linearly with token length on modern GPUs.
>
> 1. Estimated Inference FLOPs Under MORE vs. Baselines
>
> Following standard FLOPs estimates for Transformer models (≈2·N FLOPs per generated token for a model with N parameters) [1], we can approximate the per-example inference cost (encoder + decoder) for Whisper. Since the encoder runs once per utterance while the decoder runs once per output token, the relative increase in FLOPs is dominated by the increase in output length caused by MORE. In typical (non-attack) conditions on LibriSpeech, Whisper produces transcriptions roughly the same length as the reference transcript – about 22 tokens on average per utterance  Based on Table 1 in the paper, MORE can induce 10× to 14× longer transcripts compared to normal outputs across different Whisper model sizes.
>
>
> Using the parameter sizes of Whisper models and the average output lengths observed in our experiments, we obtain the following per-example FLOPs (values rounded; FLOPs in billions) on Table 1’s setting on LibriSpeech dataset:
>
>
> | Model  | Params (M) | Baseline tokens (avg) | MORE tokens (avg) | Baseline FLOPs (G) | MORE FLOPs (G) | × Increase |
> |--------|-----------:|----------------------:|------------------:|-------------------:|---------------:|-----------:|
> | Tiny   | 39         | 22                   | 296              | 1.7               | 23.1          | 13.5×     |
> | Base   | 74         | 22                   | 300              | 3.3               | 44.4          | 13.6×     |
> | Small  | 244        | 22                   | 214              | 10.7              | 104.4         | 9.7×      |
> | Medium | 769        | 22                   | 234              | 33.8              | 359.9         | 10.6×     |
> | Large  | 1550       | 22                   | 301              | 68.2              | 933.1         | 13.7×     |
>
>
> Here we approximate:
> - FLOPs per token ≈ 2 · N_params,
> - Total FLOPs per example ≈ FLOPs_encoder + (#tokens) · 2·N_params,
> and we use the empirical baseline vs. MORE token lengths from our experiments.
>
>
> These estimates show that, across all Whisper sizes, MORE increases **per-example inference compute by roughly an order of magnitude** (≈9–14×), purely by forcing the model to generate much longer, repetitive transcripts. This quantifies the **efficiency vulnerability**: MORE does not just degrade accuracy, but also inflates the FLOPs required for inference, threatening the real-time and resource efficiency of ASR deployments.
>
>
> [1] Transformer FLOPs, Adam Casson, https://www.adamcasson.com/posts/transformer-flops
>
>
> 2. Inference Time of ASR Models
> Specifically, we profile the inference time of the Whisper-Large model as a function of the number of output tokens. The plot can be found in the APPX. C. The inference time increases almost linearly with the output length. Moreover, Whisper-Large pads all inputs shorter than 30 seconds to a fixed 30-second window before processing, so utterances shorter than 30 seconds incur identical encoding time, making the output length the only factor that determines the overall resource consumption. These observations support our motivation to maximize the output length in order to induce the greatest possible waste of computational resources.

---

> ### Author Response · Authors · 2025-11-21
> **Response to question (5) (6)**
>
> Q5: Thank you for the comment. Xu et al. (2022) does not propose loop attacks or repetition failures; it analyzes natural repetition in text generation and is not an adversarial attack work. We cite it only as motivation for why transformer models exhibit repetition dynamics. SlothSpeech remains the only prior ASR efficiency-targeting attack. MORE differs from simply combining an adversarial loss with an EOS penalty because the REDO objective induces structured, periodic repetition that EOS suppression alone cannot produce. Our accuracy-initialization step and REDO mechanism jointly create controlled, large-scale repetition, which is why MORE consistently outperforms EOS-only methods like SlothSpeech in our experiments.
>
> Q6: Thank you for the question. We did not evaluate decoding-time defenses such as repetition penalties or anti-loop heuristics, as our focus is on analyzing model-level vulnerabilities rather than proposing or benchmarking defenses. Many such decoding constraints modify the inference procedure and are not standard across ASR deployments, and their effectiveness can vary widely depending on implementation. Evaluating defense strategies would require a separate and extensive study, which falls outside the scope of this work. We view investigating decoding-time or training-time defenses against multi-objective efficiency attacks as an interesting direction for future work.

---

### Official Review · Reviewer_xABV · 2025-10-28

**Soundness:** 2
**Presentation:** 2
**Contribution:** 2
**Rating:** 4
**Confidence:** 3

**Summary:**

The paper proposes an attack on automatic speech recognition (ASR) models, targeting both accuracy (ie, aiming to poison model accuracy) and sequence length (trying to produce longer sequences) at the same time. The method is a training time attack, where the attack is done hierarchically (first optimize for accuracy loss, then for producing longer sequence length), with the proposed sequence length loss including two components: i) try to suppress end-of-sentence token probability, and ii) try to encourage sentence repetition, which is a known problem that occurs also in ASR models trained without any attacks. The proposed method is tested with several Whisper family models on LibriSpeech & LJ-Speech datasets, and benchmarked against several existing adversarial attack methods.

**Strengths:**

1) Leveraging a naturally occurring failure mode for the attack seems like a nice idea.

2) The paper is mostly clear and easy to read (although see Questions for some commenting on this).

3) The proposed method is benchmarked against multiple existing methods, and the results also include a good ablation study on the individual components.

**Weaknesses:**

1) The reporting of experimental results should be improved (see Questions for details).

2) There is no implementation/source code available (although according to the Reproducibility statement it might be released later).

3) Eg lines 107-111: I do not find the argument for the need to combine the accuracy and sequence length attacks too convincing; any working accuracy degradation attack will make a given system unusable even if it is lighting fast. To me, the interesting part here is the possibility of having more effective attacks against robust (whether due to intrinsic properties of the model, cf eg Shah et al. 2025: Speech robust bench, or possible due to some defense method) models via a multiobjective attack. Also, lines 120-123: this makes very little sense to me, consider rephrasing.

4) There are no experiments nor even any discussion on how existing defense methods work against the proposed attack

**Questions:**

## Questions and comments for the authors, in decreasing order of importance:

1) Please include some variation measure, eg, standard error of the mean for all the results.

2) Please clarify how each of the hyperparameters were tuned (including for the baselines, and were these tuned separately for each method or how)?

3) How does the computational complexity of the proposed method compare with the existing methods?

4) Some steps in the proposed method are presented in a bit overly complex and jarring way, especially the repulsion step: as far as I can tell, this is basically just doing standard adversarial attack to lower the utility of the model via gradient-based optimization with simple modified loss, but this is somehow written in a grandiose way, including by calling it repulsion for some reason. I think it would improve the paper if you present this step as what it is, ie, basically a standard utility attack step (where you could probably also use any existing reasonable utility attack without affecting the results too much).

5) Tables 1-3: I do not understand how bolding is used, please make this explicit, or better yet, just highlight the best (and preferably also mark all methods eg within 1 standard error of the best)

---

> ### Author Response · Authors · 2025-11-21
> **Response to question (1) and weakness (1)**
>
> Thank you for the suggestion. We agree that variation measures such as standard errors can be informative in some evaluation settings. However, our work follows the standard reporting format used in recent adversarial attack studies on ASR and other sequence models, where results are typically reported as averaged metrics over a fixed evaluation set without per-utterance variance statistics. This is consistent with prior work on ASR accuracy attacks [1][2] and efficiency attacks [3].
>
> [1] Recent Improvements of ASR Models in the Face of Adversarial Attacks
>
> [2] Muting Whisper: A Universal Acoustic Adversarial Attack on Speech Foundation Models
>
> [3] SlothSpeech: Denial-of-service Attack Against Speech Recognition Models

---

> ### Author Response · Authors · 2025-11-21
> **Response to question (2)**
>
> Thank you for the question. To clarify, we did not perform hyperparameter tuning for MORE or for any of the baseline methods. All baseline attacks were run using their standard configurations as described in their original papers, which is the common practice in recent ASR adversarial attack studies. This ensures fair comparison without introducing method-specific optimization bias.
>
> For MORE, the hyperparameters  were selected based on general stability considerations and kept fixed across all experiments, models, and datasets. Our goal is to demonstrate the effectiveness of the proposed multi-objective design under consistent settings rather than to optimize performance through method-specific tuning.
>
> Because no hyperparameter–specific tuning was applied to any method including baselines, the comparison reflects the intrinsic behavior of each attack rather than differences arising from tuning effort. This aligns with evaluation protocols in prior ASR adversarial attack works, where attacks are typically evaluated using uniform or default parameters across models and datasets.

---

> ### Author Response · Authors · 2025-11-21
> **Response to question (3)**
>
> Thank you for the question. We treat computational complexity as a secondary aspect of the work, since the primary aim of this paper is to identify and analyze previously unaddressed efficiency vulnerabilities in ASR models rather than to optimize or benchmark attack runtime. To maintain transparency, we have included a complexity discussion for MORE in **Appendix B** and added **a FLOPs-based efficiency analysis in Appendix C**. These sections clarify how the hierarchical optimization and REDO mechanism operate, but they are not intended to position computational cost as a key contribution.
>
> Existing ASR attack papers rarely report detailed computational complexity or FLOPs measurements for their methods, and comparable information is not available for the baseline attacks we evaluate. Most baselines rely on standard PGD-style iterations whose computational behavior is well established in the literature. As a result, a direct complexity comparison is not feasible.
>
> Overall, the analysis included in the appendix serves to give readers a clear view of how MORE behaves, while the central contribution of the paper remains the demonstration of a new multi-objective attack framework that exposes an unexplored class of weaknesses in ASR systems.

---

> ### Author Response · Authors · 2025-11-21
> **Response to question (4) (5)**
>
> Q4: Thank you for the helpful comment. We agree that the accuracy-oriented part of MORE relies on a standard gradient-based utility attack that reduces the likelihood of the ground-truth transcript. The purpose of this step is not to introduce a new accuracy attack, but to create an initial misalignment in the decoding trajectory so that the efficiency-oriented objective can subsequently operate in a stable manner. Our use of the term “repulsion” was intended to convey this role of pushing the model away from its correct decoding path, rather than to imply a fundamentally new mechanism.
>
> We appreciate the reviewer’s point that presenting this step in a more direct way would improve clarity. The multi-objective contribution of MORE does not depend on the novelty of the accuracy component, and existing utility attacks could indeed be substituted without materially changing the overall framework. We have revised the text to describe this stage more clearly as a standard accuracy-degradation step used to initialize the attack, and to avoid unnecessarily elaborate terminology, and **the revised text is highlighted in blue on page 4** of the updated manuscript.
>
> Q5: Only the best performances are shown in bold.

---

> ### Author Response · Authors · 2025-11-21
> **Response to weakness (2) (3)**
>
> Q2: We will definitely release the full implementation upon acceptance to ensure complete reproducibility.
>
> Q3: Thank you for the valuable feedback. We appreciate the reviewer’s point that a sufficiently strong accuracy-degradation attack can already render an ASR system unusable. Our goal in combining accuracy and sequence-length objectives is not to suggest that efficiency degradation alone is required for practical disruption, but rather to expose a broader range of vulnerabilities in modern ASR models.
>
> Recent large-scale ASR models, including Whisper, demonstrate improved robustness to traditional accuracy-only attacks. In such cases, combining accuracy degradation with controlled manipulation of decoding dynamics can produce stronger and more reliable failures, especially for models that remain accurate but are sensitive to termination behavior and repetition loops. This perspective aligns with the reviewer’s suggestion that multi-objective attacks may be particularly valuable when targeting robust or defended systems.
>
> Regarding lines 120–123, we agree that the original phrasing was unclear. Our intention was to highlight that inducing incorrect and excessively long transcriptions reveals structural weaknesses in the decoding process, not to suggest a privacy-enhancing purpose. We have revised the text as shown below, and the same changes are highlighted in blue in the manuscript.
>
> **Revised lines 107–111**
> However, most existing approaches focus only on accuracy robustness and overlook vulnerabilities in inference efficiency, which can be exploited through decoding manipulation. SlothSpeech represents the only prior efficiency-focused attack in ASR, but it does not jointly consider accuracy degradation or structured repetition, limiting its ability to assess multi-dimensional robustness.
>
> **Revised lines 120–123**
>  By inducing incorrect and excessively long transcriptions, MORE exposes decoding weaknesses that are not revealed by accuracy-only attacks, offering a more comprehensive view of ASR vulnerability.

---

> ### Author Response · Authors · 2025-11-21
> **Response to weakness (4)**
>
> Thank you for the comment. Our work focuses on analyzing and exposing multi-objective vulnerabilities in ASR models, rather than proposing or evaluating defense mechanisms. As such, the scope of the paper is aligned with prior attack-focused research in ICLR, which typically aims to characterize model weaknesses under controlled threat settings rather than benchmark defenses.
>
> Existing defense methods for ASR primarily target accuracy-based adversarial perturbations and are not designed to address sequence-length manipulation or structured repetition, which are central to our efficiency-oriented objective. A comprehensive evaluation of defenses, many of which substantially modify model training, decoding, or input preprocessing, would require a separate and extensive study that goes beyond the goals of this work.
>
> We agree that examining how current defenses respond to multi-objective attacks such as MORE is an important direction for future work. However, since this paper is an attack and robustness analysis study rather than a defense paper, we focus on characterizing the vulnerabilities of existing ASR models rather than evaluating the full landscape of defensive techniques.

---

### Official Review · Reviewer_tS9e · 2025-11-01

**Soundness:** 3
**Presentation:** 3
**Contribution:** 2
**Rating:** 4
**Confidence:** 3

**Summary:**

This paper introduces MORE, a multi-objective adversarial attack framework targeting both attack efficacy and inference efficiency in automatic speech recognition systems. The method combines a hierarchical repulsion, anchoring optimization scheme: first degrading transcription accuracy and then prolonging decoding through EOS suppression with the repetitive encouragement doubling objective, which induces structured repetition in generated text. Experiments across two benchmarks show that MORE significantly increases both word error rate and output length, outperforming accuracy-only (standard attacks PGD, FGSM variants). Ablations confirm each component’s contribution, and complexity analysis plus ethical discussion round out the work.

**Strengths:**

The joint focus on accuracy and efficiency robustness is original and well-motivated.

Across five Whisper models, the proposed MORE method consistently yields longer, incorrect outputs (strong attack efficacy), clearly outperforming baselines.

The authors discuss potential misuse and propose mitigations, which strengthen the paper’s responsibility stance.

The appendix rigorously analyzes computational cost, showing depth of understanding.

**Weaknesses:**

The hierarchical optimization's convergence or general properties are not analyzed mathematically. More theoretical analyses should be included.

If I understand correctly, using output length as a proxy for computational efficiency is reasonable but somewhat coarse. Can authors provide some more efficiency analyses?

Only Whisper-based ASR models are tested. The paper should include the test of other architectures.

There is no black-box or transfer evaluation against closed models or API systems. It seems that all attacks are white-box.

**Questions:**

How sensitive is the proposed MORE method to hyperparameters like doubling period or the weighting between losses?

How would the approach perform on non-English or noisy real-world speech data?

Would a reinforcement-learning formulation offer a better balance between the two objectives?

---

> ### Author Response · Authors · 2025-11-21
> **Response to weakness (1) for hierarchical optimization**
>
> We thank the reviewer for pointing out the need for more theoretical analysis of the hierarchical optimization. We address this concern in two ways:
>
> 1.Clarifying existing analysis (**Appendix B**).
> Our current draft already includes a formal complexity analysis of MORE in Appendix B, where we characterize how the attack’s computational cost scales with model depth, width, and output length. In the revised version, we have: (1) explicitly reference Appendix B (Complexity Analysis) in the main text when introducing the hierarchical optimization, so that readers can easily find this theoretical treatment. (2) Clarify that this analysis applies to both stages of MORE (repulsion and anchoring), and formally ties the REDO-induced length growth to the asymptotic increase in inference complexity.
>
> 2. Together with the new FLOPs-based efficiency analysis (added in response to weakness 2), these additions provide both a theoretical characterization (in terms of complexity and stationary structure of the objective) and an empirical view of the optimization dynamics of MORE.

---

> ### Author Response · Authors · 2025-11-21
> **Response to weakness (2) for efficiency analyses**
>
> We thank the reviewer for their insight in pointing out the need to provide more efficiency analyses according to FLOPs. We add these analyses to **Appendix C**. Specifically, we compare the estimated inference FLOPs of MORE and the baselines. We also measure the actual inference time of samples, which better reflects real-world computational efficiency, as shown in **Figure 2 in Appendix C on page 18**.
>
> 1. Estimated Inference FLOPs Under MORE vs. Baselines
>
> Following standard FLOPs estimates for Transformer models (≈2·N FLOPs per generated token for a model with N parameters) [1], we can approximate the per-example inference cost (encoder + decoder) for Whisper. Since the encoder runs once per utterance while the decoder runs once per output token, the relative increase in FLOPs is dominated by the increase in output length caused by MORE. In typical (non-attack) conditions on LibriSpeech, Whisper produces transcriptions roughly the same length as the reference transcript – about 22 tokens on average per utterance. Based on Table 1 in the paper, MORE can induce 10× to 14× longer transcripts compared to normal outputs across different Whisper model sizes.
>
>
> Using the parameter sizes of Whisper models and the average output lengths observed in our experiments, we obtain the following per-example FLOPs (values rounded; FLOPs in billions) on Table 1’s setting on LibriSpeech dataset:
>
>
> | Model  | Params (M) | Baseline tokens (avg) | MORE tokens (avg) | Baseline FLOPs (G) | MORE FLOPs (G) | × Increase |
> |--------|-----------:|----------------------:|------------------:|-------------------:|---------------:|-----------:|
> | Tiny   | 39         | 22                   | 296              | 1.7               | 23.1          | 13.5×     |
> | Base   | 74         | 22                   | 300              | 3.3               | 44.4          | 13.6×     |
> | Small  | 244        | 22                   | 214              | 10.7              | 104.4         | 9.7×      |
> | Medium | 769        | 22                   | 234              | 33.8              | 359.9         | 10.6×     |
> | Large  | 1550       | 22                   | 301              | 68.2              | 933.1         | 13.7×     |
>
>
> Here we approximate:
> - FLOPs per token ≈ 2 · N_params,
> - Total FLOPs per example ≈ FLOPs_encoder + (#tokens) · 2·N_params,
> and we use the empirical baseline vs. MORE token lengths from our experiments.
>
> These estimates show that, across all Whisper sizes, MORE increases **per-example inference compute by roughly an order of magnitude** (≈9–14×), purely by forcing the model to generate much longer, repetitive transcripts. This quantifies the **efficiency vulnerability**: MORE does not just degrade accuracy, but also inflates the FLOPs required for inference, threatening the real-time and resource efficiency of ASR deployments.
>
> [1] Transformer FLOPs, Adam Casson, https://www.adamcasson.com/posts/transformer-flops
>
>
> 2. **Inference Time of ASR Models in Figure 2 on page 18**
>
> Specifically we have added Figure 2, we profile the inference time of the Whisper-Large model as a function of the number of output tokens. The plot can be found in the Appendix C on page 18. The inference time increases almost linearly with the output length. Moreover, Whisper-Large pads all inputs shorter than 30 seconds to a fixed 30-second window before processing, so utterances shorter than 30 seconds incur identical encoding time, making the output length the only factor that determines the overall resource consumption. These observations support our motivation to maximize the output length in order to induce the greatest possible waste of computational resources.

---

> > ### Author Response · Authors · 2025-11-21
> > **Response to weakness (3)**
> >
> > We focus on Whisper because it is one of the most widely used ASR models, both for standalone recognition and as the speech encoder in many multimodal LLMs. Demonstrating successful attacks on Whisper across accuracy and efficiency therefore provides meaningful insights into ASR robustness. We appreciate the suggestion to include additional architectures and plan to extend our evaluation to other models in future work.

---

> > > ### Author Response · Authors · 2025-11-21
> > > **Response to weakness (4)**
> > >
> > > Our work focuses on dual-objective attacks, targeting both accuracy and efficiency in a white-box setting. We consider strengthening white-box attacks along these two objectives to be an essential and challenging step toward advancing ASR attack research. While black-box or transfer evaluations against closed models are certainly valuable, they fall outside the scope of this paper. We acknowledge their importance and plan to investigate transferability to closed-source systems in future work.

---

> > > > ### Author Response · Authors · 2025-11-21
> > > > **Response to question (1)**
> > > >
> > > > Thank you for the insightful question. MORE is designed so that its key hyperparameters play interpretable roles and do not require delicate tuning to obtain the reported results.
> > > >
> > > > Regarding the efficiency-stage loss weighting, the second stage combines the EOS suppression loss and the REDO loss. In our implementation, we use a simple additive combination with equal weights and do not tune these coefficients separately for different models or datasets. Conceptually, these two terms are not competing objectives: EOS suppression encourages the model not to terminate, while REDO shapes the continuation into a structured repetition. Both terms therefore act in the same direction (longer outputs) but with complementary effects (whether to continue, and how to continue), which reduces the need for fine-grained balancing. In our experiments, this straightforward combination already yields stable optimization behavior and strong efficiency degradation, suggesting that MORE does not rely on a fragile or highly tuned choice of loss weights. For the doubling period in REDO, this hyperparameter controls how frequently the repeated target is updated rather than defining a strict optimal operating point. Smaller values of the period make the model adapt to repeated segments more aggressively, whereas larger values spread repetition updates over longer horizons. In practice, we use a moderate fixed period that balances stability and output elongation; the qualitative behavior of producing long, repetitive transcriptions remains consistent for reasonable choices around this setting. The efficiency gains observed in our results do not arise from carefully optimizing this period per model, but from the underlying design of REDO and the hierarchical attack framework.
> > > >
> > > > Overall, while MORE does have hyperparameters such as the doubling period and the relative scaling of efficiency components, the method is constructed so that these parameters have clear roles and do not need to be finely tuned to specific architectures or datasets to obtain effective joint accuracy and efficiency attacks.

---

> > > > > ### Author Response · Authors · 2025-11-21
> > > > > **Response to question (2)**
> > > > >
> > > > > Thank you for the question. Although our experiments focus on English benchmark datasets, the proposed attack is built on model-level mechanisms, namely EOS suppression, repetition reinforcement through REDO, and gradient-based perturbation of the autoregressive decoding process, that are not language-dependent. Whisper models are trained on large multilingual corpora and share a unified tokenization and decoding process across languages. Since MORE perturbs the acoustic input and manipulates decoding dynamics rather than relying on language-specific features, its underlying principles are expected to transfer to other languages supported by the model.
> > > > >
> > > > > Regarding noisy real-world speech, our method does not assume clean input conditions and already demonstrates effectiveness at lower SNR levels, which emulate more challenging acoustic environments. The components of MORE operate on model gradients and decoding behavior, and these mechanisms remain present even when the input contains background noise. As shown in our 30 dB and 35 dB evaluations, the attack continues to degrade both accuracy and efficiency even in noisier conditions, suggesting resilience across noise levels.

---

> > > > > > ### Author Response · Authors · 2025-11-21
> > > > > > **Response to question (3)**
> > > > > >
> > > > > > Thank you for the insightful comment. Reinforcement-learning (RL)–based formulations are indeed a possible direction for multi-objective sequence optimization. However, in the context of adversarial attacks on ASR models, RL introduces several practical challenges that make it less suitable than our current gradient-based approach.
> > > > > >
> > > > > > First, RL typically requires defining reward functions that reflect both accuracy and efficiency degradation. Designing stable and well-shaped rewards for autoregressive ASR decoding is non-trivial, and small changes in reward scaling can lead to unstable or unpredictable behavior. In contrast, MORE leverages direct gradient signals from the model, enabling precise and efficient optimization without the need for reward engineering.
> > > > > >
> > > > > > Second, RL approaches generally incur much higher computational cost, since they rely on sampling multiple sequences through the decoder during training. This can be prohibitively expensive for large ASR models such as Whisper. MORE avoids this overhead by using deterministic, differentiable objectives (EOS suppression and REDO) that directly shape token-level decoding behavior.
> > > > > >
> > > > > > Finally, RL does not inherently simplify the trade-off between objectives. In fact, RL often requires tuning additional hyperparameters (e.g., reward weights, discount factors, exploration strategies), potentially increasing sensitivity rather than reducing it. The hierarchical design in MORE provides a clearer and more stable separation between accuracy degradation and efficiency manipulation, which serves our attack setting effectively.
> > > > > >
> > > > > > In summary, while RL is an interesting alternative, it introduces additional complexities and overhead without clear benefits for adversarial ASR attacks. Our gradient-based hierarchical formulation allows for a more direct, stable, and computationally efficient way to jointly influence accuracy and efficiency.

---

### Official Review · Reviewer_P3t3 · 2025-11-01

**Soundness:** 2
**Presentation:** 2
**Contribution:** 2
**Rating:** 2
**Confidence:** 4

**Summary:**

Large-scale automatic speech recognition (ASR) models like Whisper are widely used, making robustness to small input changes crucial. Prior studies mainly address accuracy loss under adversarial attacks, robustness with respect to efficiency remains largely unexplored. To fill this gap, the authors propose MORE (Multi-Objective Repetitive Doubling Encouragement), an attack that jointly reduces recognition accuracy and inference efficiency. Using a hierarchical repulsion–anchoring mechanism, MORE sequentially optimizes these objectives. Its REDO (Repetitive Encouragement Doubling Objective) promotes duplicative text generation, degrading accuracy while doubling sequence length. Experiments show MORE yields longer, error-prone transcriptions at higher computational cost, exposing new multi-objective vulnerabilities in ASR models.

**Strengths:**

The experiments consider multiple DL models and datasets, with seemingly more rigorous evaluation methodologies than many previous papers in this area.

**Weaknesses:**

(1) Even though the paper claims to focus on robustness with respect to efficiency against attacks, the experimental results appear to primarily emphasize accuracy or, in some cases, the trade-off instead. However, accuracy has already been extensively explored in prior works (Raina et al., 2024; Raina & Gales, 2024; Olivier & Raj, 2022b; Madry et al., 2018a; Dong et al., 2018; Wang & He, 2021; Gao et al., 2024). Therefore, the overall contribution of this paper seems quite limited.

(2) Lack of a Clear Threat Model. It is unclear what assumptions the paper makes regarding the attack setting. Is this a black-box or a white-box attack? What knowledge does the attacker have about the victim model and dataset? These aspects need to be clearly specified in the paper. I recommend that the authors add a dedicated subsection titled “Threat Model” to explicitly describe the assumptions and settings, so that readers can better understand the scope and validity of the proposed method.

(3) Insufficient Experimental Evaluation. The experimental analysis in the paper is not sufficient. In particular, the transferability of adversarial attacks is a crucial aspect when evaluating the effectiveness of an attack method. However, the paper does not include any experiments assessing transferability. It remains unclear whether an attack generated on one model can be successfully transferred to and applied against another model. Including such an evaluation would significantly strengthen the empirical validation of the proposed approach.

(4) Finally, the writing needs significant improvement. In critical parts of the paper, it is hard to tell what the authors did in terms of experimentation and analysis, or what motivated the choices they made.

**Questions:**

Please refer to my comments for more details.

---

> ### Author Response · Authors · 2025-11-21
> **Response to weakness (4)**
>
> Thank you for the helpful feedback regarding clarity. We acknowledge that some parts of the manuscript required improved explanation of the experimental procedure, the motivation behind our design choices, and the flow of analysis. In the updated version, we have included clearer descriptions of the technical motivations, the objective design, and the optimization process, and we have refined the structure of the experimental section to make the evaluation easier to follow. These updates, highlighted in blue in the revised manuscript, improve readability while preserving the original technical contributions.

---

> ### Author Response · Authors · 2025-11-21
> **Response to weakness (3)**
>
> We appreciate that transferability is an important dimension in the adversarial ML literature. However, our paper specifically focuses on **dual-objective white-box attacks targeting accuracy and efficiency**, which constitute a substantial departure from prior accuracy-only or transfer-focused studies.
>
> Efficiency-oriented attacks require direct optimization of the model-specific decoding dynamics (e.g., encouraging repetitive generation via REDO), and such mechanisms inherently rely on access to gradients and decoder behavior. For this reason, transferability is not the central objective of our study, and we view it as an important but orthogonal research direction.
>
> We agree that investigating cross-model transferability of efficiency-based attacks would be valuable future work, and we will explicitly state this in the revision. We thank the reviewer for raising this point.

---

> ### Author Response · Authors · 2025-11-21
> **Response to weakness (2)**
>
> We agree that clarifying the threat model improves readability. MORE operates under a white-box threat model, where the attacker has access to model parameters and gradients. The victim ASR model and the corresponding training distribution are assumed to be observable, consistent with standard settings in gradient-based adversarial attack research.
>
> We thank the reviewer for the suggestion and have revised the paper to include a dedicated “Threat Model” subsection clearly stating: (1) the attack is white-box and (2) model parameters are accessible.
>
> This below clarification on page 6 will help readers better understand the scope and assumptions of our work.
>
> “**Threat Model**. We conduct white-box attacks with full access to the model on five Whisper-family models (Radford et al., 2023), including Whisper-tiny, Whisper-base, Whisper-small, Whisper-medium, and Whisper-large, all obtained from HuggingFace. To benchmark the proposed MORE approach, we compare it against five strong white-box attack baselines: PGD (Olivier & Raj, 2022b), MI-FGSM (Dong et al., 2018), VMI-FGSM (Wang & He, 2021), the speech-aware gradient optimization (SAGO) method (Gao et al., 2024), and SlothSpeech (Haque et al., 2023).”

---

> ### Author Response · Authors · 2025-11-21
> **Response to weakness (1)**
>
> We appreciate the reviewer’s observation. We would like to clarify that the goal of our work is to introduce multi-objective attack against ASR systems that leads to both accuracy and efficiency degradation. To the best of our knowledge, **MORE is the first attack designed to jointly degrade accuracy and inference efficiency**, exposing multi-objective vulnerabilities that existing methods do not capture.
>
> Although accuracy results are included for completeness and comparison, our primary goal, and the key novelty of the work, is to (i) design an explicit mechanism that induces repetitive over-generation, (ii) demonstrate that this mechanism significantly increases inference cost, and (iii) show that modern ASR architectures are susceptible to this novel class of efficiency attacks.
>
> Our experiments highlight this dual objective. As shown in Table 1 and Table 2, MORE achieves state-of-the-art efficiency-attack performance across two datasets, significantly increasing output length compared with five strong baselines. At the same time, MORE maintains competitive accuracy-attack performance, ensuring that the attack remains effective in both dimensions rather than optimizing one at the expense of the other. The case study in Table 4 further illustrates how MORE simultaneously degrades accuracy and inflates output length, with the repetitive text patterns clearly demonstrating the success of the efficiency attack.
>
> Our contributions therefore go beyond traditional accuracy-based adversarial work and highlight a distinct vulnerability dimension that, we believe, is important for practical ASR deployment.

---

### Meta-Review · Area_Chair_enKF · 2026-01-02

**Summary:**

The key reviewers’ concerns can be summarized as follows: (i) evaluation is limited to encoder-decoder autoregressive architectures; (ii) there is no black-box or transfer evaluation against closed models or API systems; (iii) transferability of adversarial attacks; (iv) lack of experiments showing how existing defense methods perform against the proposed attack.

**Reviewer Concerns:**

The authors were able to address some of the concerns related to experimental settings (e.g., hyperparameters) and the efficacy of the proposed solution in real-world noisy speech conditions. However, other  fundamental concerns still remain after the rebuttal phase. Specifically: The approach is tested only against a single model, Whisper, which limits the experimental analysis and makes the claim of a comprehensive ASR robustness evaluation overreaching. Furthermore, the authors do not explain how the method would prevent repetition or efficiency degradation under different ASR decoding strategies, such as CTC. With respect to the transferability of adversarial attacks, the authors’ response,  namely, that transferability is not the central objective of the study,  suggests that, although the proposed approach is interesting, it is still not fully mature.

**Reviewer Scores:**

P3t3 might not have changed their score.

tS9e might not have changed their score.

xABV might have increased their score by +1.0.

v3bB might not have changed their score.

---

### Decision · Program_Chairs · 2026-01-26

Reject